# Hsc70 promotes anti-tumor immunity by targeting PD-L1 for lysosomal degradation

Xiaoyan Xu[1,13], Tingxue Xie[1,2,13], Mengxin Zhou[1,2,13], Yaqin Sun[2,13], Fengqi Wang[1,2,13], Yanan Tian[1,2], Ziyan Chen[3], Yanqi Xie[4], Ronghai Wu[5], Xufeng Cen[1], Jichun Zhou [6], Tingjun Hou [7], Lei Zhang[8], Chaoyang Huang[9], Qingwei Zhao[10], Dongrui Wang[11,12] ✉ & Hongguang Xia [1,2] ✉

Immune checkpoint inhibition targeting the PD-1/PD-L1 pathway has become a powerful clinical strategy for treating cancer, but its efficacy is complicated by various resistance mechanisms. One of the reasons for the resistance is the internalization and recycling of PD-L1 itself upon antibody binding. The inhibition of lysosome-mediated degradation of PD-L1 is critical for preserving the amount of PD-L1 recycling back to the cell membrane. In this study, we find that Hsc70 promotes PD-L1 degradation through the endosome-lysosome pathway and reduces PD-L1 recycling to the cell membrane. This effect is dependent on Hsc70-PD-L1 binding which inhibits the CMTM6-PD-L1 interaction. We further identify an Hsp90α/β inhibitor, AUY-922, which induces Hsc70 expression and PD-L1 lysosomal degradation. Either Hsc70 overexpression or AUY-922 treatment can reduce PD-L1 expression, inhibit tumor growth and promote anti-tumor immunity in female mice; AUY-922 can further enhance the anti-tumor efficacy of anti-PD-L1 and anti-CTLA4 treatment. Our study elucidates a molecular mechanism of Hsc70-mediated PD-L1 lysosomal degradation and provides a target and therapeutic strategies for tumor immunotherapy.

Immune suppression and dysfunction is one of the hallmarks of cancer, while reinvigorating the anti-tumor immune responses has been a rapidly-evolving field for cancer treatment[1–3]. PD-1/PD-L1 immune checkpoint inhibition (ICI) is a promising clinical therapy for some types of cancer[4,5], but a significant proportion of patients still developed resistance[6,7]. PD-L1 tends to internalize after the binding of antibody, during which some of it is lysosome-mediated degradation[8]. However, most of the internalized PD-L1 can be stored in recycling endosomes and repopulated to the cell membrane[9], mediating resistance to ICI treatment targeting PD-L1[10,11]. Therefore,

[1]Research Center of Clinical Pharmacy of The First Affiliated Hospital & Liangzhu Laboratory, Zhejiang University School of Medicine, Hangzhou, China. [2]Department of Biochemistry, Zhejiang University School of Medicine, Hangzhou, China. [3]Department of Urology, the First Affiliated Hospital, Zhejiang University School of Medicine, Hangzhou, China. [4]Department of Urology, the Second Affiliated Hospital, Zhejiang University School of Medicine, Hangzhou, China. [5]Hangzhou PhecdaMed Co.Ltd, 2626 Yuhangtang Road, Hangzhou, China. [6]Department of Surgical Oncology, Affiliated Sir Run Run Shaw Hospital, Zhejiang University School of Medicine, Hangzhou, China. [7]College of Pharmaceutical Sciences, Zhejiang University, Hangzhou, China. [8]Department of Cardiology/Health Management Center, The First Affiliated Hospital, Zhejiang University School of Medicine, Hangzhou, China. [9]Department of Cardiology, the First Affiliated Hospital, Zhejiang University School of Medicine, Hangzhou, China. [10]Department of Clinical Pharmacy, the First Affiliated Hospital, Zhejiang University School of Medicine, Hangzhou, China. [11]Bone Marrow Transplantation Center, the First Affiliated Hospital, and Liangzhu Laboratory, Zhejiang University School of Medicine, Hangzhou, China. [12]Zhejiang Province Engineering Laboratory for Stem Cell and Immunity Therapy, Hangzhou, China. [13]These authors contributed equally: Xiaoyan Xu, Tingxue Xie, Mengxin Zhou, Yaqin Sun, Fengqi Wang. ✉e-mail: dongrui-wang@zju.edu.cn; hongguangxia@zju.edu.cn

promoting lysosomal degradation and inhibiting the recycling of PD-L1 is a promising approach to overcome the resistance to ICI therapy[12]. CMTM6 as a ubiquitously-expressed protein that has been shown to aid the recycling of PD-L1 back to the plasma membrane, and depletion of CMTM6 promoted PD-L1 degradation through lysosomes[9,13], however, the lysosomal degradation mechanism of PD-L1 is still not clear. Therefore, investigating the mechanism and regulation of PD-L1 lysosomal degradation can provide new ideas for improving immune efficacy.

Most research on PD-L1 degradation is focused on the proteasome pathway, and the study of PD-L1 degradation in the endosomal lysosomal pathway is scare. Hsc70 (heat shock protein family A (Hsp70) member 8) is a cytoplasmic chaperone protein which plays a crucial role in endosomal microautophagy (eMI) and chaperone-mediated autophagy (CMA)[14]. About 40% of mammalian proteins contain KFERQ-like motif, which can serve as the substrate of Hsc70[15,16]. Hsc70 directly delivers cargo proteins to lysosomes by interacting with Lamp2a through CMA[17] or interacts with negatively charged phosphatidylserine via its carboxy-terminal lid domain, leading to cytosolic cargo internalization into endosomes through eMI[18,19]. Although CMA and eMI have similar motifs, their substrates are not completely overlapped, when proteins with similar KFERQ motifs are in a semi-aggregated state, they form high molecular weight complexes or cannot be unfolded and degraded by CMA, but they can still be degraded by eMI[14]. Post translational modifications could also determine whether the proteins are routed to eMI or CMA. One recent study showed that acetylated Tau reduces it degradation by CMA, while promoting its engagement by eMI[20]. Recently, eMI has attracted attention because of its underlying molecular mechanism and potential biological functions. eMI as selective autophagy, plays an important role in the degradation of biomacromolecules, but its research is relatively lacking.

In this work, we use mass spectrometry to identify Hsc70 as one of the major PD-L1-interacting proteins. Mechanistic studies reveal that overexpression of Hsc70 promotes the degradation of PD-L1 through the endosome-lysosome pathway. Hsc70 competes with CMTM6 to bind with PD-L1, promoting lysosomal degradation of PD-L1 and reducing the recycling of PD-L1 to cell membrane. We further screen for an inhibitor of Hsp90, AUY-922, which is found to prompt PD-L1 degradation through endosome and lysosome by enhancing Hsc70 levels. Both Hsc70 overexpression and AUY-922 treatment inhibit tumor growth and promote anti-tumor T cell responses; AUY-922 enhances the anti-tumor effect of aPD-L1 and aCTLA4 treatment, therefore providing an approach for developing future cancer immunotherapy.

## Results

### Hsc70 promotes lysosomal degradation of PD-L1

ICI using anti-PD-L1 antibody (aPD-L1) can trigger PD-L1 internalization and recycling. In this study, we confirmed this process in 4T1 breast cancer cells, showing that PD-L1 expression on the cell membrane significantly decreased after short-term aPD-L1 treatment (24 h), but rebounded as the treatment prolonged (Supplementary Fig. 1A). To find out potential mediators of PD-L1 lysosomal degradation, PD-L1 was immunoprecipitated and subjected to mass spectrometry analysis. Some high-confidence interacting proteins were identified (Fig. 1A). To further investigate the clinical relevance of these proteins, we analyzed their protein levels in relation to overall survival in Breast cancer patients using Kaplan-Meier Plotter analysis (Supplementary Fig. 1B). Among these proteins, only HSPA8 and FUS were included and showed positive correlations with overall survival. Our goal is to overcome the resistance to immunotherapy, so we compared the correlation between HSPA8 or FUS expression and overall survival under immunotherapy conditions of anti-PD-L1, as shown in Supplementary Fig. 1C, only HSPA8 showed a positive correlation with overall

survival. Therefore, we chose Hsc70 (HSPA8) for subsequent detailed investigations. To investigate whether and how Hsc70 regulates PD-L1 protein expression, we overexpressed Hsc70 in MCF-7 and PANC1 cells. We discovered that Hsc70 significantly decreased the total abundance (Fig. 1B, Supplementary Fig. 1D), as well as the surface expression (Fig. 1C, Supplementary Fig. 1E) of PD-L1. Real-time PCR results showed that Hsc70 overexpression did not alter the transcription levels of PD-L1 (Supplementary Fig. 1F), and Hsc70 overexpression also simultaneously degraded exogenous expressed PD-L1 level (Supplementary Fig. 1G), indicating that Hsc70 regulates PD-L1 expression at the protein level.

To confirm that Hsc70-induced PD-L1 downregulation was dependent on lysosome-mediated protein degradation, we treated Hsc70-overexpressing cells with inhibitors of lysosome (Leupeptin +NH₄Cl or E64D), proteasome (MG132) or autophagy (3-MA). We found that only the inhibition of lysosome function can restore PD-L1 expression in the context of Hsc70 overexpression (Fig. 1D, Supplementary Fig. 1H). Therefore, our results suggested that PD-L1 degradation promoted by Hsc70 is lysosome-dependent.

Chaperone-mediated autophagy (CMA) is a selective form of autophagy in which proteins are recognized by Hsc70 and delivered to the lysosomal receptor Lamp2a, resulting in translocation into lysosomes and degradation. Lamp2a is considered as the rate limiting protein of CMA[21], but either overexpression or knockdown of Lamp2a in MCF-7 and PANC1 cells had no effect on total PD-L1 levels and surface PD-L1 expression (Fig. 1E–H, Supplementary Fig. 1I–K). Further, Spautin1 and AC220 treatment has been known to induce CMA by degrading the critical negative regulator HK2[22,23]. Treating MCF-7 cells with Spautin-1 and AC220 was able to degrade HK2 (Fig. 1I), but the total abundance or surface expression of PD-L1 were not affected (Fig. 1I, J). Consistent results were found in PANC1 cells and the IFN-γ stimulated U937 cells (Supplementary Fig. 1L–N). These experiments suggest that PD-L1 degradation induced by Hsc70 is dependent on lysosome but not on CMA.

### Hsc70 induces PD-L1 degradation via eMI

Hsc70 has been reported as a key chaperone protein in both CMA and eMI pathways[24]. Therefore, we next tested whether PD-L1 is degraded via eMI. The ESCRT-I complex, TSG101, plays an essential role in the formation of MVB (multivesicular body) and facilitates the transport of cytoplasmic proteins to the endosome[16]. VPS4 is another critical component of eMI and is the regulatory ATPase of ESCRT-III[25]. To determine whether these key regulators of eMI are involved in the Hsc70-induced PD-L1 degradation, we knocked down TSG101 or VPS4 in MCF-7 and PANC1 cells, and found that knockdown of either factor blocked the degradation of total PD-L1 induced by Hsc70 (Fig. 2A, C; Supplementary Fig. 2A, B) and resulted in a recovery of membrane PD-L1 expression (Fig. 2B, D; Supplementary Fig. 2C, D). Further, the cholesterol transport inhibitor U18666A has been reported to block MVB dynamics and inhibit eMI[16,26] and U18666A treatment in MCF-7 and PANC1 cells also blocked PD-L1 degradation (Fig. 2E; Supplementary Fig. 2E) and recovered surface PD-L1 expression (Fig. 2F; Supplementary Fig. 2F). These results suggested that Hsc70-induced PD-L1 degradation is dependent on eMI.

The binding of Hsc70 carboxy-terminal lid domain with phosphatidylserine on the late endosome membrane is essential for eMI[18,19]. Hsc70-3KA is an Hsc70 mutant form on its C-terminal lysine residues which disrupts the binding to phosphatidylserine[16]. We found that overexpression of Hsc70-3KA did not result in PD-L1 degradation or surface PD-L1 reduction as compared to the wild-type protein (Hsc70-WT) (Fig. 2G, H; Supplementary Fig. 2G, H). Consistently, overexpression of Hsc70-WT increased the co-localization of PD-L1 with late endosomes (RAB7A, Fig. 2I; Supplementary Fig. 2I) and lysosomes (LAMP1, Fig. 2J; Supplementary Fig. 2J), while decreasing the co-localization of PD-L1 with recycling endosomes (RAB11, Fig. 2K; Supplementary Fig. 2K) that facilitates its return to the cell membrane[9].

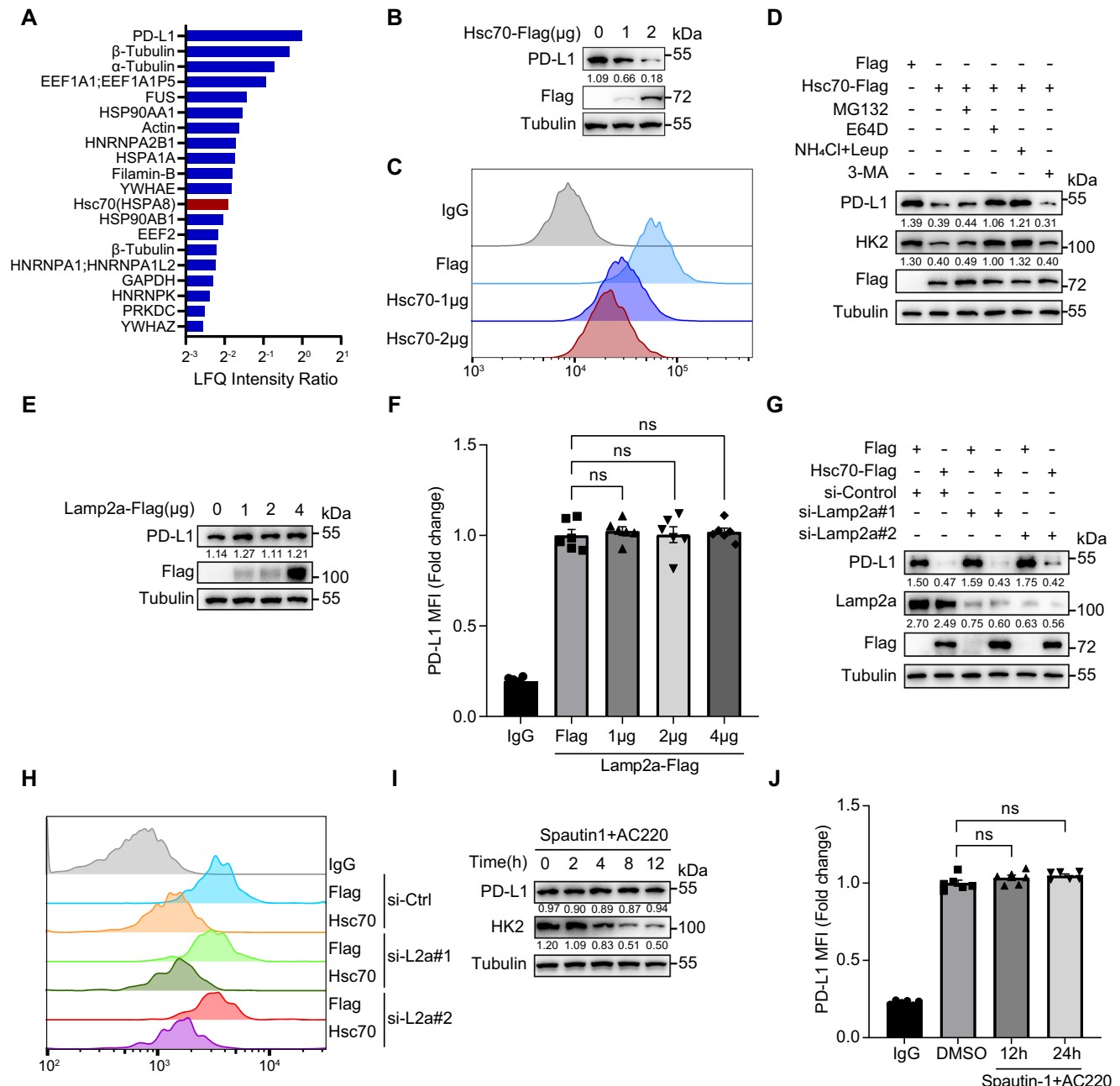

**Fig. 1 | Hsc70 promotes PD-L1 degradation through lysosome. A** HEK293T cells were transfected with Flag or PD-L1-Flag for 24 h, Flag-tagged PD-L1 were trypsin-digested and protein identification and quantitation was through LC-MS/MS. The top-twenty PD-L1-interacting proteins were identified by Mass spectrometry. **B, C** MCF-7 cells were transfected with 1 μg and 2 μg Hsc70-Flag for 24 h, and total PD-L1 levels were detected by western blotting (**B**), fluorescence of PD-L1 on the surface of cell membrane was analyzed by flow cytometry (**C**). **D** MCF-7 cells were transfected with Flag or Hsc70-Flag for 12 h, treated with or without MG132 (10 μM), NH₄Cl (20 mM), Leupeptin (100 nM), 3-MA (5 mM), E-64D (10 μM) for another 12 h. Cell lysates were immunoblotted with indicated antibodies. **E, F** MCF-7 cells were transfected with 1 μg, 2 μg or 4 μg Lamp2a-Flag for 24 h. Indicated protein levels were detected by western blotting (**E**), fluorescence of PD-L1 on the surface of cell membrane was analyzed by flow cytometry, $n = 6$, ns: not significance (**F**). **G, H** MCF-7 cells were transfected with siRNAs of Lamp2a for 36 h, transfected with or without Hsc70-Flag for another 24 h. Cell lysates were immunoblotted with indicated antibodies (**G**), fluorescence of PD-L1 on the surface of cell membrane was analyzed by flow cytometry (**H**). **I, J** MCF-7 cells were treated with Spautin-1 (10 μM) and AC220 (2 μM) for 0, 2, 4, 8, and 12 h. Cell lysates were immunoblotted with indicated antibodies (**I**), fluorescence of PD-L1 on the surface of cell membrane was analyzed by flow cytometry, $n = 6$, ns: not significance (**J**). For (**C, F, H, J**), MCF-7 cells were seeded in 48-well plate with 6 replicates per group and subjected to the corresponding treatment, repeated independently three times and similar results were obtained. Data shown in (**B, D, E, G, I**) were repeated independently three times with similar results. Data represent Mean ± SEM, for (**F**) and (**J**) data, two-sided with adjustment of Tukey's multiple comparisons, one-way ANOVA. The numbers under blots represent the value (the ratio to Tubulin) of grayscale quantification. Source data are provided as a Source data file.

However, these effects were not observed when Hsc70-3KA was overexpressed (Fig. 2I–K; Supplementary Fig. 2I–K). These findings further support that Hsc70 promotes PD-L1 degradation via the endosome and lysosome pathway.

## Hsc70 competes with CMTM6 for binding and regulation of PD-L1

CMTM6 has been reported to inhibit PD-L1 degradation through lysosome[9], leading us to investigate how Hsc70 and CMTM6

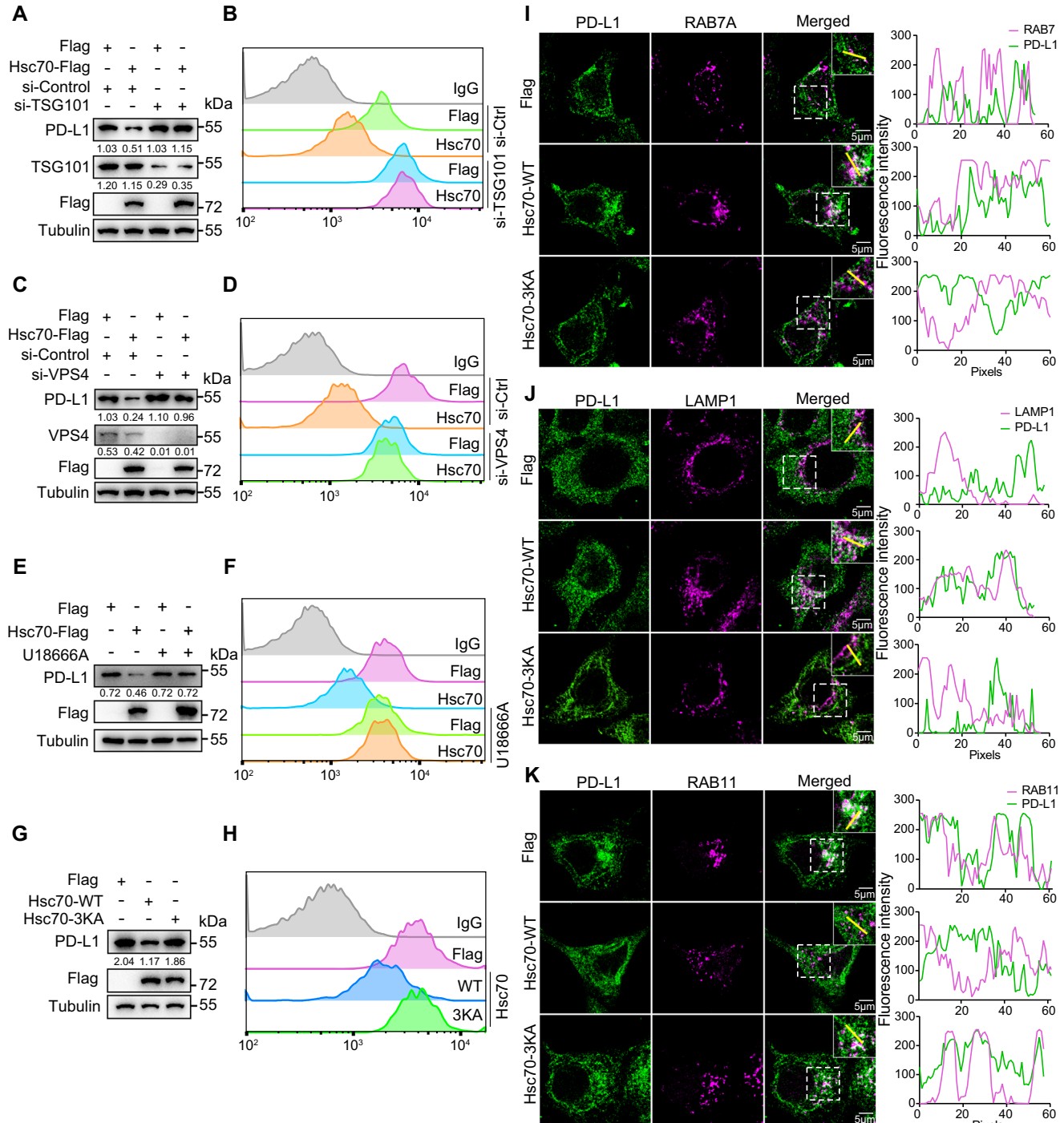

**Fig. 2 | Hsc70 induces PD-L1 degradation via eMI. A, B** MCF-7 cells were transfected with siRNAs of TSG101 for 36 h, transfected with or without Hsc70-Flag for another 24 h. Cell lysates were immunoblotted with indicated antibodies (**A**), fluorescence of PD-L1 on the surface of cell membrane was analyzed by flow cytometry (**B**). **C, D** MCF-7 cells were transfected with siRNAs of VPS4 for 36 h, transfected with or without Hsc70-Flag for another 24 h. Cell lysates were immunoblotted with indicated antibodies (**C**), fluorescence of PD-L1 on the surface of cell membrane was analyzed by flow cytometry (**D**). **E, F** MCF-7 cells were transfected with Flag or Hsc70-Flag for 12 h, treated with or without U18666A (5 µg/mL) for another 12 h. PD-L1 levels were detected by western blotting (**E**), fluorescence of PD-L1 on the surface of cell membrane was analyzed by flow cytometry (**F**). **G, H** MCF-7 cells were transfected with Hsc70-WT-Flag or Hsc70-3KA-Flag plasmids for 24 h. PD-L1 protein levels were detected by western blotting (**G**), fluorescence of PD-L1

on the surface of cell membrane was analyzed by flow cytometry (**H**). **I–K** MCF-7 cells were transfected with Flag, Hsc70-WT-Flag or Hsc70-3KA-Flag for 8 h, the co-localization between RAB7A and PD-L1 (**I**), the co-localization between LAMP1 and PD-L1 (**J**), the co-localization between RAB11 and PD-L1 (**K**) were done using immunofluorescence and confocal microscopy. Scale bar, 5 µm. The intensity profiles along the yellow line are plotted in the middle panels, with the colocalizing sites marked by white. For (**B**, **D**, **F**, **H**), MCF-7 cells were seeded in 48-well plate with 6 replicates per group and subjected to the corresponding treatment, repeated independently three times and similar results were obtained. Data shown in (**A**), (**C**), (**E**), (**G**), and (**I–K**) were repeated independently three times and similar results were obtained. The numbers under blots represent the value (the ratio to Tubulin) of grayscale quantification. Source data are provided as a Source data file.

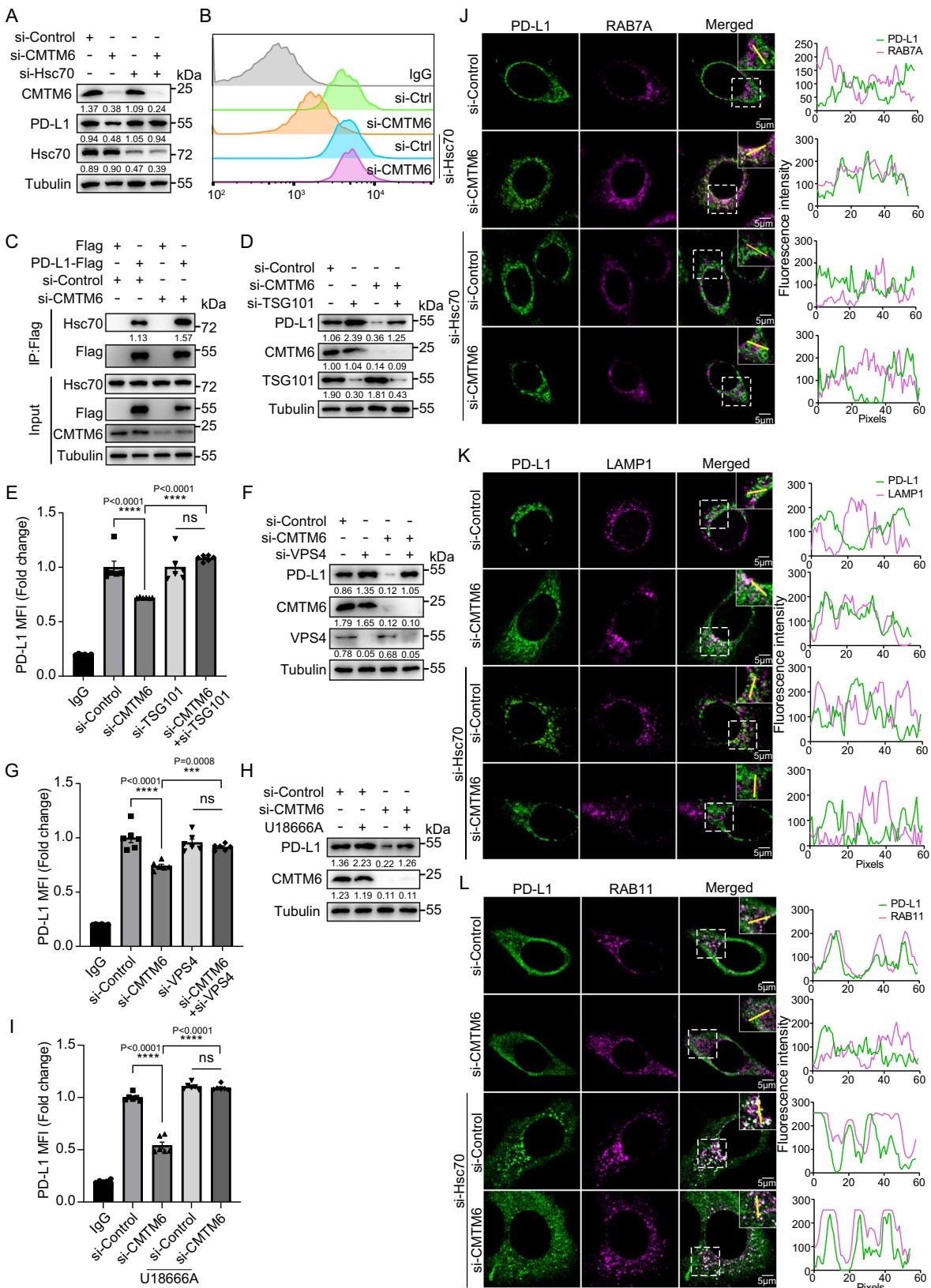

cooperatively affect PD-L1 protein expression. We first found that neither overexpression nor knockdown of Hsc70 altered CMTM6 levels (Supplementary Fig. 3A, B). Also, knockdown of CMTM6 did not change Hsc70 levels despite the significant reduction of total and surface PD-L1 (Supplementary Fig. 3C, D). However, while CMTM6 knockdown reduced PD-L1 protein expression, an additional

knockdown of Hsc70 led to a recovery of total PD-L1 levels (Fig. 3A) and surface PD-L1 expression (Fig. 3B, Supplementary Fig. 3E), suggesting that CMTM6 knockdown-mediated PD-L1 degradation is dependent on Hsc70.

Co-immunoprecipitation assays showed that the interaction between exogenous PD-L1 and endogenous Hsc70 was enhanced in

**Fig. 3 | Hsc70 competes with CMTM6 for binding and regulation of PD-L1.**
**A**, **B** MCF-7 cells were transfected with siRNA of Hsc70 and CMTM6 for 60 h, PD-L1 levels were detected by western blotting (**A**), fluorescence of PD-L1 on the surface of cell membrane was analyzed by flow cytometry (**B**). **C** HEK293T cells were transfected with siRNA of CMTM6 for 48 h, then transfected with PD-L1-Flag for another 12 h, the interaction between Hsc70 and PD-L1 was detected by immuno-precipitation. **D**, **E** MCF-7 cells were transfected with siRNAs of CMTM6 and TSG101 for 60 h, cell lysates were immunoblotted with indicated antibodies (**D**), fluores-cence of PD-L1 on the surface of cell membrane was analyzed by flow cytometry, $n = 6$ (**E**). **F**, **G** MCF-7 cells were transfected with siRNAs of CMTM6 and VPS4 for 60 h, cell lysates were immunoblotted with indicated antibodies (**F**), fluorescence of PD-L1 on the surface of cell membrane was analyzed by flow cytometry, $n = 6$ (**G**). **H**, **I** MCF-7 cells were transfected with siRNA of CMTM6 for 48 h, treated with or without U18666A (5 μg/mL) for another 12 h, PD-L1 levels were detected by western

blotting (**H**), fluorescence of PD-L1 on the surface of cell membrane was analyzed by flow cytometry, $n = 6$ (**I**). **J**–**L** MCF-7 cells were transfected with indicated siRNAs for 36 h and the co-localization between RAB7A and PD-L1 (**J**), the co-localization between LAMP1 and PD-L1 (**K**), the co-localization between RAB11 and PD-L1 (**L**) were done using immunofluorescence and confocal microscopy. Scale bar, 5 μm. The intensity profiles along the yellow line are plotted in the middle panels, with the colocalizing sites marked by white. For (**B**, **E**, **G**, **I**), MCF-7 cells were seeded in 48-well plate with 6 replicates per group and subjected to the corresponding treat-ment, repeated independently three times and similar results were obtained. Data shown in (**A**), (**C**), (**D**), (**F**), (**H**), and (**J**–**L**) were repeated independently three times with similar results. Data represent Mean ± SEM, for (**E**, **G**, **I**) data, two-sided with adjustment of Tukey's multiple comparisons, one-way ANOVA, *P* value is indicated in the graph. The numbers under blots represent the value (the ratio to Tubulin/Flag) of grayscale quantification. Source data are provided as a Source data file.

CMTM6-knockdown 293 T cells (Fig. 3C), indicating a competitive binding with PD-L1 between Hsc70 and CMTM6. To test this hypothesis, we constructed various truncated proteins of PD-L1 (Supplementary Fig. 3F). Immunoprecipitation assay revealed that the main binding site of Hsc70 was located in the position of aa 161-241 (Supplementary Fig. 3G) which overlaps with the binding site of CMTM6 and PD-L1 (Supplementary Fig. 3H), providing mechanistic explanation why knocking down CMTM6 enhanced the interaction between Hsc70 and PD-L1.

To evaluate the functional consequence of competitive binding between Hsc70 and CMTM6, we next tested whether CMTM6-knockdown-mediated PD-L1 degradation is dependent on eMI. We knocked down TSG101 or VPS4 to inhibit eMI, showing a recovery of total PD-L1 (Fig. 3D, F) and membrane PD-L1 (Fig. 3E, G) in the context of CMTM6 knockdown. Consistently, U18666A also prevented the degradation of total PD-L1 (Fig. 3H) and membrane PD-L1 (Fig. 3I) mediated by CMTM6 knockdown. Furthermore, Immuno-fluorescence assays showed that CMTM6 knockdown enhanced the interaction between PD-L1 and RAB7A (marker of late endosome), which was disrupted with simultaneous knockdown of CMTM6 and Hsc70 (Fig. 3J; Supplementary Fig. 3I). Similar results were observed in the co-localization of PD-L1 and lysosomes (LAMP1) (Fig. 3K; Supplementary Fig. 3J). Knocking down CMTM6 reduced the co-localization of PD-L1 and recycling endosomes (RAB11), while knocking down Hsc70 did not affect this interaction (Fig. 3L; Supplementary Fig. 3K). Taken together, these findings suggest that CMTM6 and Hsc70 competitively bind to PD-L1, and CMTM6 knockdown enhanced the interaction between Hsc70 and PD-L1 to promote the degradation of PD-L1 through eMI.

## TFG is involved in endosome-lysosome degradation of PD-L1
Since Hsc70-induced PD-L1 degradation is mainly dependent on the interaction between Hsc70 and phosphatidylserine in late endosomes, we compared the interacting proteins between Hsc70-WT and Hsc70-3KA using mass spectrometry to discover additional mediators of this process (Fig. 4A). The results showed that the COPII-mediated vesicle transport pathway was highly enriched in the Hsc70-WT group (Fig. 4B). Among the proteins in this pathway, TFG, which is known as a SEC16-interacting protein to coordinate the distribution of COPII transport carriers[27,28], was highly enriched in Hsc70-WT and was in the top-ranked hit of PD-L1 interacting proteins (Fig. 4A, C). Subsequent immunoprecipitation tests confirmed that the interaction between TFG and Hsc70-WT was significantly stronger than that of Hsc70-3KA (Fig. 4D). Given the recently reported study that impaired TFG function compromises the transport of at least a subset of endosomal cargoes[29], we speculated that TFG plays a role in the process of Hsc70-mediated PD-L1 lysosomal degradation. Therefore, we overexpressed TFG in MCF-7 cells and found that it promoted the degradation of PD-L1 (Fig. 4E). In contrast, knockdown of TFG blocked the degradation of PD-L1 without affecting Hsc70 expression

(Fig. 4F). Knockdown of TFG also inhibited the degradation of total PD-L1 (Fig. 4G), and increased the level of surface PD-L1 (Fig. 4H) in the context of Hsc70 expression. Similarly, knockdown TFG results in reduced co-localization of PD-L1 with late endosomes (RAB7A, Fig. 4I; Supplementary Fig. 4A) and lysosomes (LAMP1, Fig. 4J; Supplementary Fig. 4B). These results suggested that TFG is involved in the lysosomal degradation of PD-L1.

## Hsc70 promotes anti-tumor immunity and controls in vivo tumor growth
The discovery that Hsc70 promotes PD-L1 degradation led us to investigate the role of Hsc70 in anti-tumor immunity. We over-expressed Hsc70-WT or Hsc70-3KA in 4T1 murine breast cancer cells, which were then inoculated into immunocompetent and immune-deficient mice (Fig. 5A). We observed a dramatic suppression of tumor growth in the Hsc70-WT group compared with the Control and Hsc70-3KA group in immunocompetent BALB/c mice. Intriguingly, the tumor-suppressive function of Hsc70-WT was not observed in immunodefi-cient nude mice (Fig. 5B–D). These findings suggest that the anti-tumor effect of Hsc70 is related to the immune responses. CD8+ T cells are critical effectors of anti-tumor immunity, eliminating cancer cells by secreting granzyme B and induce apoptosis in target cells. Therefore, we examined the intratumoral CD8+ T cell population and activated (GzmB+) CD8+ cytotoxic T cells by flow cytometry. Our results showed that the levels of CD8+ and activated CD8+ cells were significantly increased in the tumors with Hsc70-WT overexpression (Fig. 5E, F). Consistent results were also observed by immunohistochemistry (Fig. 5G). Furthermore, analyses of harvested tumor cells showed that overexpression of Hsc70-WT significantly reduced PD-L1 levels (Fig. 5H). These results suggest that Hsc70 can downregulate PD-L1, enhance T cell effector function and reduce tumorigenesis in vivo.

To assess whether the functional impact of Hsc70 overexpression is mediated through PD-L1, we examined the individual and combined effects of PD-L1 knockdown and Hsc70 overexpression on tumor growth. We established stable transgenic 4T1 cell lines that knocked down PD-L1 with or without overexpressed wild-type Hsc70 and inoculated them into immunocompetent BALB/c mice (Supplementary Fig. 5A). Our findings revealed that Hsc70 overexpression (Hsc70-OE) or PD-L1 knockdown (PD-L1-KD) effectively suppressed tumor growth, while the combination of both (Hsc70-OE + PD-L1-KD) did not further enhance the inhibitory effect (Supplementary Fig. 5B–D). These results suggested that Hsc70 overexpression may not have an additive effect on PD-L1 knockdown. Our findings also demonstrated a marked increase in the levels of CD8+ and activated CD8+ cells in tumors with PD-L1 knockdown or Hsc70 overexpression (Supplementary Fig. 5E–G). PD-L1 levels in the harvested tumor cells showed a dramatic decrease after Hsc70 overexpression (Supplementary Fig. 5H). Collectively, these findings suggest that the effect of Hsc70 overexpression on anti-tumor T cell response is mainly mediated through PD-L1.

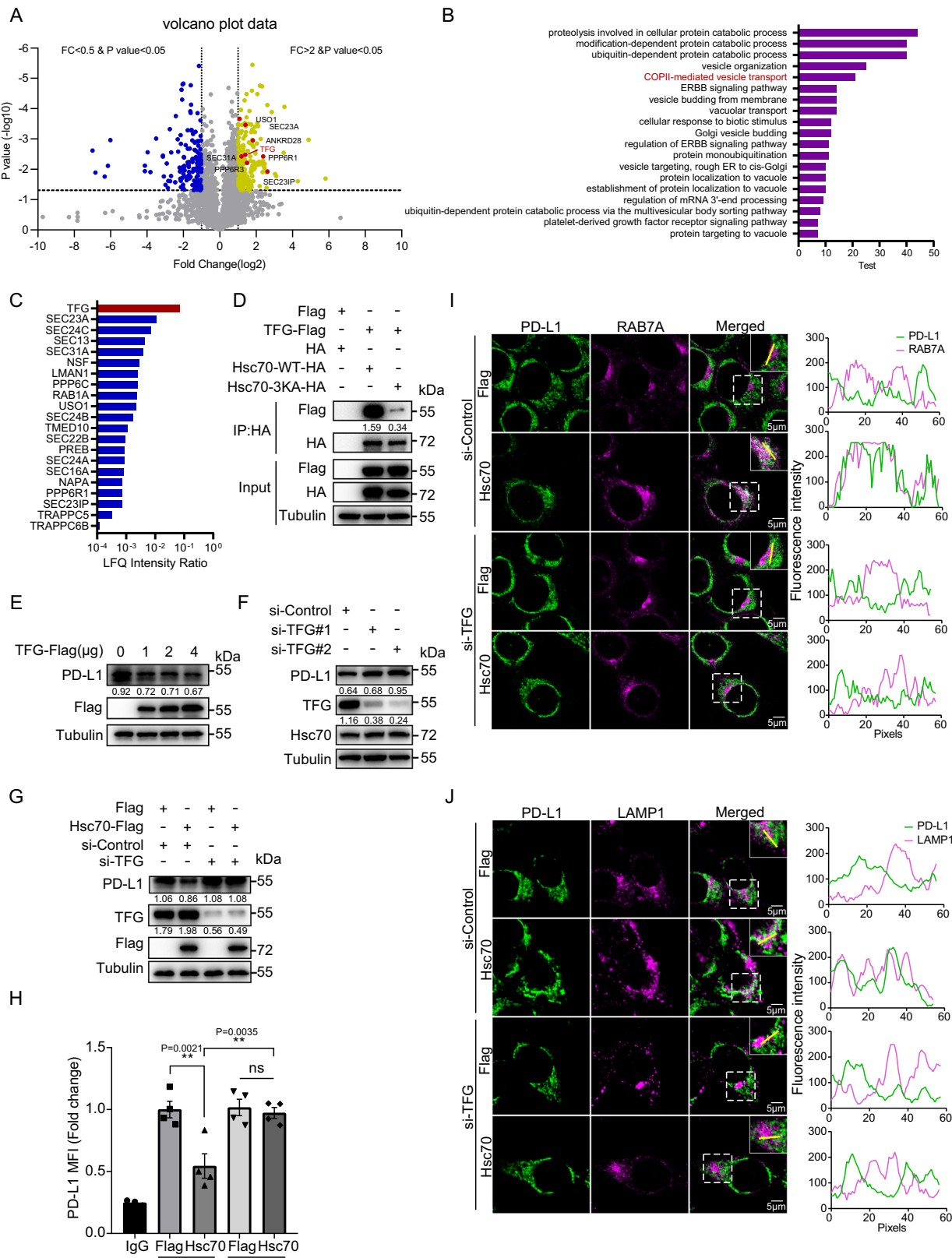

### AUY-922 promotes PD-L1 degradation through Hsc70-mediated eMI

Through the analysis of our previous high-throughput screening results[30], we found that an Hsp90 inhibitor, AUY-922 (Fig. 6A), can reduce the level of PD-L1 (Fig. 6B, C). Interestingly, we observed that AUY-922 treatment significantly enhanced the levels of Hsc70 while

degrading PD-L1 (Fig. 6B). The results were also seen in PANC1 and U937 cells (Supplementary Fig. 6A–D). To verify whether AUY-922 promotes PD-L1 degradation through Hsc70, we knocked down Hsc70 after AUY-922 treatment in MCF-7 cells. The results showed that knockdown of Hsc70 prevented the degradation of total PD-L1 (Supplementary Fig. 6E) and resulted in an increase of surface PD-L1

**Fig. 4 | TFG is involved in endosome-lysosome degradation of PD-L1. A** Volcano map of significantly differentially interaction protein of Hsc70-WT compared with Hsc70-3KA. The significantly down-regulated proteins are marked with blue (FC < 0.5 and $p$ < 0.05), the significantly upregulated proteins are marked with yellow (FC > 2 and $p$ < 0.05), the proteins with no difference are gray. COPII-mediated vesicle transport proteins are marked in red. FC and $P$ values were tested by unpaired $t$-test, with three mass spectrometry replicates/group. **B** Differently interaction proteins of Hsc70-WT compared with Hsc70-3KA under biological process classification GO function enrichment diagram. **C** Ranking of COPII-mediated vesicle transport proteins in PD-L1 interacting proteins. **D** HEK293T cells were transfected with indicated plasmids for 24 h, the interaction between TFG and Hsc70 was detected by immunoprecipitation. **E** MCF-7 cells were transfected with 1 μg, 2 μg and 4 μg TFG-Flag for 24 h, PD-L1 levels were detected by western blotting. **F** MCF-7 cells were transfected with siRNA of TFG for 60 h, indicated protein levels were detected by western blotting. **G, H** MCF-7 cells were transfected with siRNA of TFG for 36 h, then transfected with or without Hsc70 for another 24 h,

indicated protein levels were detected by western blotting (**G**), fluorescence of PD-L1 on the surface of cell membrane was analyzed by flow cytometry, data represent Mean ± SEM, two-sided with adjustment of Tukey's multiple comparisons, one-way ANOVA test. $n$ = 4, $P$ value is indicated in the graph (**H**). **I, J** MCF-7 cells were transfected with siRNA of TFG for 48 h, then transfected with or without Hsc70 for another 8 h, the co-localization between RAB7A and PD-L1 (**I**), the co-localization between LAMP1 and PD-L1 (**J**) were analyzed by immunofluorescence and confocal microscopy. Scale bar, 5 μm. The intensity profiles along the yellow line are plotted in the middle panels, with the colocalizing sites marked by white. For (**H**), MCF-7 cells were seeded in 24-well plate with 4 replicates per group and subjected to the corresponding treatment, repeated independently three times and similar results were obtained. Data shown in (**D–G**) and (**I, J**) were repeated independently three times with similar results. The numbers under blots represent the value (the ratio to Tubulin/HA) of grayscale quantification. Source data are provided as a Source data file.

---

expression (Supplementary Fig. 6F). We also found that the degradation of PD-L1 induced by AUY-922 was dependent on lysosome but not proteasome or macroautophagy (Fig. 6D; Supplementary Fig. 6G). The degradation of PD-L1 did not depend on Lamp2a (Fig. 6E, F; Supplementary Fig. 6H), suggesting that AUY-922 induced the degradation of PD-L1 was not through CMA. We then speculated that eMI is a critical downstream mechanism of AUY-922-induced Hsc70 that promotes PD-L1 degradation.

To test this hypothesis, we knocked down TSG101 or VPS4 in MCF-7 cells and treated these cells with AUY-922. We found that knockdown of TSG101 or VPS4 also prevented the degradation of total PD-L1 (Supplementary Fig. 6I, J) and led to an increase of PD-L1 surface expression (Fig. 6G, H) in AUY-922-treated cells. Consistently, in AUY-922-treated MCF-7 cells, adding the eMI inhibitor U18666A prevented total PD-L1 degradation (Supplementary Fig. 6K) and induced surface PD-L1 expression (Fig. 6I). Similar results were also observed in PANC1 cells (Supplementary Fig. 6L–N). Furthermore, AUY-922 treatment increased the co-localization of PD-L1 with late endosomes (RAB7A, Fig. 6J; Supplementary Fig. 6O) and lysosomes (LAMP1, Fig. 6K; Supplementary Fig. 6P), but decreased the co-localization of PD-L1 with recycling endosomes (RAB11, Fig. 6L; Supplementary Fig. 6Q). These results suggest that AUY-922 promotes the degradation of PD-L1 through eMI.

### AUY-922 promotes anti-tumor immunity and enhances the therapeutic effect of aPD-L1 or aCTLA4 ICI

To confirm the in vivo anti-tumor function of AUY-922, we established breast cancer mouse models using 4T1 cells inoculated to immunocompetent and immunodeficient mice, which were then treated with AUY-922 (Supplementary Fig. 7A). The results revealed that AUY-922 treatment significantly impeded tumor growth in immunocompetent BALB/c mice, evident by the decreased tumor volume and weight (Supplementary Fig. 7B–D). However, no such effect was observed in immunodeficient nude mice (Supplementary Fig. 7B–D). Moreover, the levels of total and activated CD8[+] cytotoxic T cells (GzmB[+]) that infiltrated the tumor microenvironment were significantly increased in the AUY-922 treated group (Supplementary Fig. 7E, F). These observations were further confirmed through immunohistochemistry (Supplementary Fig. 7G). Consistent with the in vitro results, AUY-922 treatment in vivo resulted in a decline in PD-L1 levels and an increase in Hsc70 levels (Supplementary Fig. 7H). These results demonstrated the in vivo anti-tumor and immune-stimulatory effect of AUY-922.

Similar to the effect of Hsc70 overexpression, AUY-922 treatment also significantly reduced the level of PD-L1 in cells treated with aPD-L1 antibody (Supplementary Fig. 8), suggesting that AUY-922 can be exploited to overcome the resistance to aPD-L1 treatment. Thus, we used the 4T1 tumor mouse model and performed aPD-L1 ICI treatment

in the presence/absence of AUY-922 (Fig. 7A). Reduction of tumor growth was observed in either AUY-922 or aPD-L1 monotherapy, but a further decrease of tumor growth was observed in the AUY-922 and aPD-L1 combinational treatment group (Fig. 7B–D). The levels of total and activated (GzmB[+]) intratumoral CD8[+] cytotoxic T cells were significantly increased upon the AUY-922/aPD-L1 combinational treatment compared to aPD-L1 monotherapy (Fig. 7E, F). These observations were further confirmed through immunohistochemistry (Fig. 7G). Furthermore, AUY-922 treatment reduced the surface expression and total protein levels of PD-L1 (Fig. 7H, I). We next evaluated the effect of AUY-922 combined with anti-CTLA4 ICI treatment (Supplementary Fig. 9A). Reduction of tumor growth was observed in either AUY-922 or anti-CTLA monotherapy, but a further decrease of tumor growth was observed in the AUY-922 and anti-CTLA4 combinational treatment group (Supplementary Fig. 9B–D). The levels of total and activated intratumoral CD8[+] cytotoxic T cells were significantly increased upon combinational treatment compared to anti-CTLA4 monotherapy (Supplementary Fig. 9E–G). Combinational treatment or AUY-922 alone reduced the surface expression of PD-L1 and upregulated Hsc70 levels (Supplementary Fig. 9H, I). Collectively, our results have suggested a potential drug candidate for combinational immunotherapy with ICI treatment.

## Discussion

Understanding the regulatory mechanism of PD-L1 is crucial for developing more cancer immunotherapy strategies. Recent studies have uncovered that after aPD-L1 treatment, the induction of intracellular PD-L1 and the subsequent re-expression to the cell membrane[31–33]. This mechanism confers resistance to ICI therapy by the re-distribution of membrane PD-L1. Hence, controlling the intracellular PD-L1 can serve as a potent anti-tumor strategy. Our study highlights that upregulation of Hsc70 competitively binds PD-L1 with CMTM6, thereby driving PD-L1 degradation via endosomes and lysosomes and reducing the amount of PD-L1 recirculating back to the cell membrane.

This mechanism was further found to enhance the effector T cell response and reduce tumor growth. Remarkably, the Hsp90 inhibitor, AUY-922, can facilitate PD-L1 degradation via lysosomes and boost anti-tumor immunity by augmenting Hsc70 expression, which can serve as a potential drug candidate for combinational immunotherapy. The anti-tumor response mediated by Hsc70 overexpression was shown in a syngeneic breast cancer mouse model.

It has been revealed that PD-L1 is predominantly degraded through two distinct pathways: proteasome and lysosome. Several studies have explored the regulation of PD-L1 proteasomal degradation, with some insights indicating that inhibiting PD-L1 glycosylation or deubiquitination promotes PD-L1 degradation via proteasomes[30,34,35]. Other research has suggested that HIP1R targets PD-L1 to lysosomal degradation and

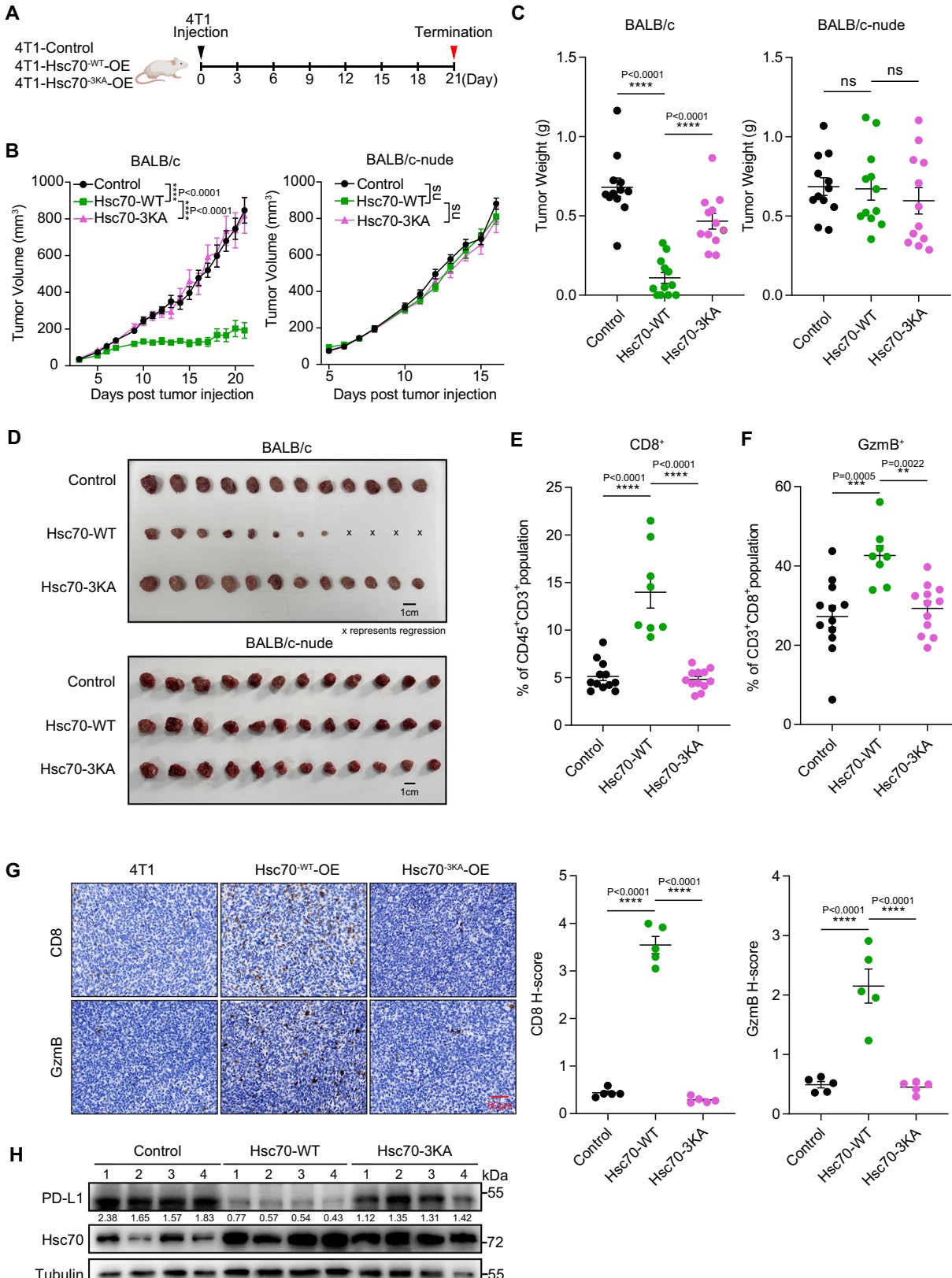

alters T cell-mediated cytotoxicity[36]. CMTM6 knockdown was also shown to promote PD-L1 degradation through lysosomes[9,13]. These results have suggested an important role of lysosomes in PD-L1 degradation and warranted further mechanistic study.

Most research on PD-L1 degradation is focused on the proteasome pathway, and the study of PD-L1 degradation in the endosomal lysosomal pathway is scare. Our results demonstrate that Hsc70 facilitates PD-L1 degradation via endosomes and lysosomes. Especially, we found that Hsc70 and CMTM6 share the binding region of PD-L1 and CMTM6-knockdown-mediated PD-L1 degradation requires Hsc70 and is achieved through endosomal microautophagy. Consistently, we also found that knocking down CMTM6 enhanced

**Fig. 5 | Hsc70 promotes anti-tumor immunity and PD-L1 degradation in vivo.**
**A** The schematic diagram of 4T1 breast cancer tumor model, created with BioRender.com, released under a Creative Commons Attribution-NonCommercial-NoDerivs 4.0 International license. **B–D** Tumor growth of Control ($n = 12$), Hsc70-WT ($n = 12$), and Hsc70-3KA ($n = 12$) group which injected indicated cells in BALB/c and BALB/c-nude mice and final tumor weights. (X represents tumor regression during the final dissection). **E, F** Flow cytometry analysis of CD8$^+$ T cells (**E**) and CD8$^+$GzmB$^+$ T cells (**F**) in tumors, Control ($n = 12$), Hsc70-WT ($n = 8$) and Hsc70-3KA ($n = 12$). **G** Immunohistochemistry analysis of CD8 and GzmB in the 4T1 tumors.

Scale bar, 50 μm. $n = 5$ (each group has 5 tumor tissues, each tissue has 3 random fields). **H** Indicated protein levels were detected by Immunoblotting with the indicated harvested tumor cells, $n = 4$ (randomly selected tumor tissues from 4 mice in each group for detection) and repeated independently two times with similar results. Data represent Mean ± SEM, for B data two-way ANOVA, for (**C**), (**E–G**) data one-way ANOVA, two-sided with adjustment of Tukey's multiple comparisons, $P$ value is indicated in the graph. The numbers under blots represent the value (the ratio to Tubulin) of grayscale quantification. Source data are provided as a Source data file.

the interaction of PD-L1 and Hsc70, which has suggested a potential mechanism of CMTM6 and Hsc70 cooperatively regulate PD-L1 lysosomal degradation.

Hsc70-mediated eMI serves as a crucial selective autophagy mode for degrading biological macromolecules[37,38], but its underlying mechanisms remain relatively obscure. Previous studies have shown that the assembly level and dynamic changes of ESCRT proteins are likely involved in the regulation of eMI[39] and the COPII vesicle-related protein TFG functions on autophagy induction[40]. TFG participates in the remodeling of the endomembrane system, and is an essential factor for protein transport in the early secretory pathway[41,42]. Recent research identified an unexpected role for TFG in trafficking of Rab4A-positive recycling endosomes specifically within axons and dendrites. Impaired TFG function compromises the transport of at least a subset of endosomal cargoes[29]. Our data suggests that TFG may participate in the degradation of PD-L1 which mediated by endosome and lysosome. The detailed mechanism of TFG-mediated degradation, including the fusion between substrate protein and endosome, and changes the expression of inner body membrane proteins, can be intriguing directions for future investigation.

Intracellular Hsc70 level is abundant and stable across different types of cells, leading to the challenges to intervene its expression for therapeutic purposes[14]. In our study, we identified that AUY-922, an inhibitor of Hsp90α/β[43] promoted the degradation of PD-L1 through eMI by enhancing the levels of Hsc70. Induced expression of heat shock proteins (Hsps) plays a central role in promoting cellular survival after environmental and physiological stress[44]. There may be functional redundancy between Hsp90α/β and Hsc70. Therefore, inhibiting the function of Hsp90α/β can be used as a strategy to enhance the expression of Hsc70. On the other hand, we found that knockdown of CMTM6 enhanced the interaction of Hsc70 and PD-L1, promoted PD-L1 endosome-lysosome degradation, suggesting that CMTM6 inhibitors also can be used as the preferred compound to enhance the function of Hsc70.

Hsc70/HSPA8 is capable of inserting into the lipid bilayer[45] and has been reported to be expressed on cell surface[46] as well as endosome membranes participating in the microautophagy[16,18,45]. In our study, we observed the consistent expression pattern of Hsc70 (Supplementary Fig. 10A, B). Therefore, we infer that endogenous-level Hsc70 plays a critical role in mediating basal levels of PD-L1 recycling and degradation. In contrast, an elevated expression of Hsc70 or an augmentation in Hsc70 levels induced by AUY-922 disrupts the fundamental PD-L1 recycling mechanism. On one hand, this results in enhanced internalization of cytoplasmic PD-L1 by Hsc70, leading to its transport to late endosomes where it undergoes degradation by lysosomes. On the other hand, Hsc70 competes with CMTM6 for binding sites on PD-L1, thus diminishing PD-L1 to recycle back to the cell membrane via CMTM6. Together, this diminished recycling reduces the interaction between PD-L1 on tumor cells and PD-1 on T cells, ultimately reactivating T cell-mediated anti-tumor immunity (Fig. 8).

In summary, our study has suggested that enhancing the interaction between Hsc70 and substrate proteins such as PD-L1 can alleviate disease progression. These results also raised the possibility that regulation of eMI by regulating the amount of protein or modification of Hsc70 could also be beneficial to other eMI-related diseases, including cancer, neurodegenerative diseases and other aging-related diseases.

## Methods
### Cell culture
HEK293T (ATCC CRL-3216), MCF-7 (ATCC HTB-22), and PANC1 (ATCC CRL-1469) cells were grown in DMEM medium (Hyclone, with L-glutamine, with 4.5 g/L glucose, without pyruvate); U937 (CTCC-001-0027, Meisen) and 4T1 (ATCC CRL-2539) cells were grown in RPMI-1640 medium (Hyclone, with L-glutamine). All cell lines used in this study were authenticated by short tandem repeat profiling. Also, all cell lines were confirmed mycoplasma free. Cells STR test reports were attached in Source Data. These media were supplemented with 10% FBS (Yeasen, 40130ES76), 1% Penicillin/Streptomycin (Gibco™). PD-L1-KD stable cell, PD-L1-KD+Hsc70-OE stable cell lines, Hsc70-WT-OE stable cell lines and Hsc70-3KA-OE stable cell lines were grown in RPMI-1640 medium (Hyclone, with L-glutamine).

### Reagents and antibody generation
The chemicals and their sources are as follows: Puromycin (#A606719), NH$_4$Cl (#A501569) from Sangon® Biotech; AUY-922 (#S1069), MG132 (#S2619), E-64D (#S7393), Leupeptin hemisulfate (#S7380), Spautin-1 (#S7888), AC220 (#S1526), 3-MA (#S2767) from TargetMol; U18666A (#662015) from Sigma; Anti-Flag (DYKDDDDK) Affinity Gel (#B23102) was purchased from Bimake. Lipofectamine™ 2000 (#1901433) and Lipofectamine ™ 3000 (#2067450) were from Invitrogen. The recombinant cytokines and their sources are as follows: Human recombinant IFNγ (#300-02) and Mouse recombinant IFNγ (#315-05) were purchased from Peprotech. The antibodies were provided as follows: HK2 (#22029-1-AP, 1:1000, Proteintech), PD-L1 (#66248-1-Ig, 1:1000, Proteintech), Hsc70 (#10654-1-AP, 1:2000, Proteintech), β-tubulin (#M1305-2, 1:5000, HUABIO), Flag-Tag (#M1403-2, 1:2000, HUABIO), HA-Tag (#0906-1, 1:2000, HUABIO), Lamp2a (#ab18528, 1:1000, Abcam), TSG101 (#ab83, 1:1000, abcam), VPS4 (#ab229806, 1:1000, abcam), CMTM6 (#HPA026980, 1:1000, Sigma), TFG (#ab156866, 1:1000, Abcam). The following antibodies were used in immunofluorescence: LAMP1 (#D4O1S, 1:100, Cell Signaling Technology), RAB7A (#E9O7E, 1:100, Cell Signaling Technology), PD-L1 (#D8T4X, 1:100, Cell Signaling Technology), RAB11 (#610656, 1:50, BD Science). The antibodies for immunohistochemistry (IHC) were used: CD8α (#98941, 1:200, Cell Signaling Technology), Granzyme B (#44153, 1:100, Cell Signaling Technology). The following antibodies for flow cytometry analysis were displayed: PE anti-human CD274 (29E.2A3) (#329706, 1:200, Biolegend), PE Mouse IgG2b, κ Isotype Ctrl (MPC-11) (#400312, 1:200, Biolegend), PE anti-mouse CD274 (MIH6) (#153612, 1:200, Biolegend), PE Rat IgG2a, λ Isotype Ctrl (G013C12) (#402304, Biolegend), Zombie Violet™ Fixable Viability Kit (#423114; 1:200; Biolegend), PerCP/Cyanine5.5 anti-mouse CD45 (#103132; 1:200; Biolegend), PE/Cyanine7 anti-mouse CD3 (#100320; 1:200; Biolegend), FITC anti-mouse CD8 (#100706; 1:200; Biolegend), APC/Cyanine7 anti-mouse CD8a (#100714; 1:200; Biolegend), APC anti-human/mouse Granzyme B (#372204; 1:200; Biolegend). The secondary antibodies for western blot were used: goat anti-mouse (#31430,1:20,000, Thermo Fisher Scientific), goat anti-rabbit (#31460, 1:20,000, Thermo Fisher Scientific). The fluorescent

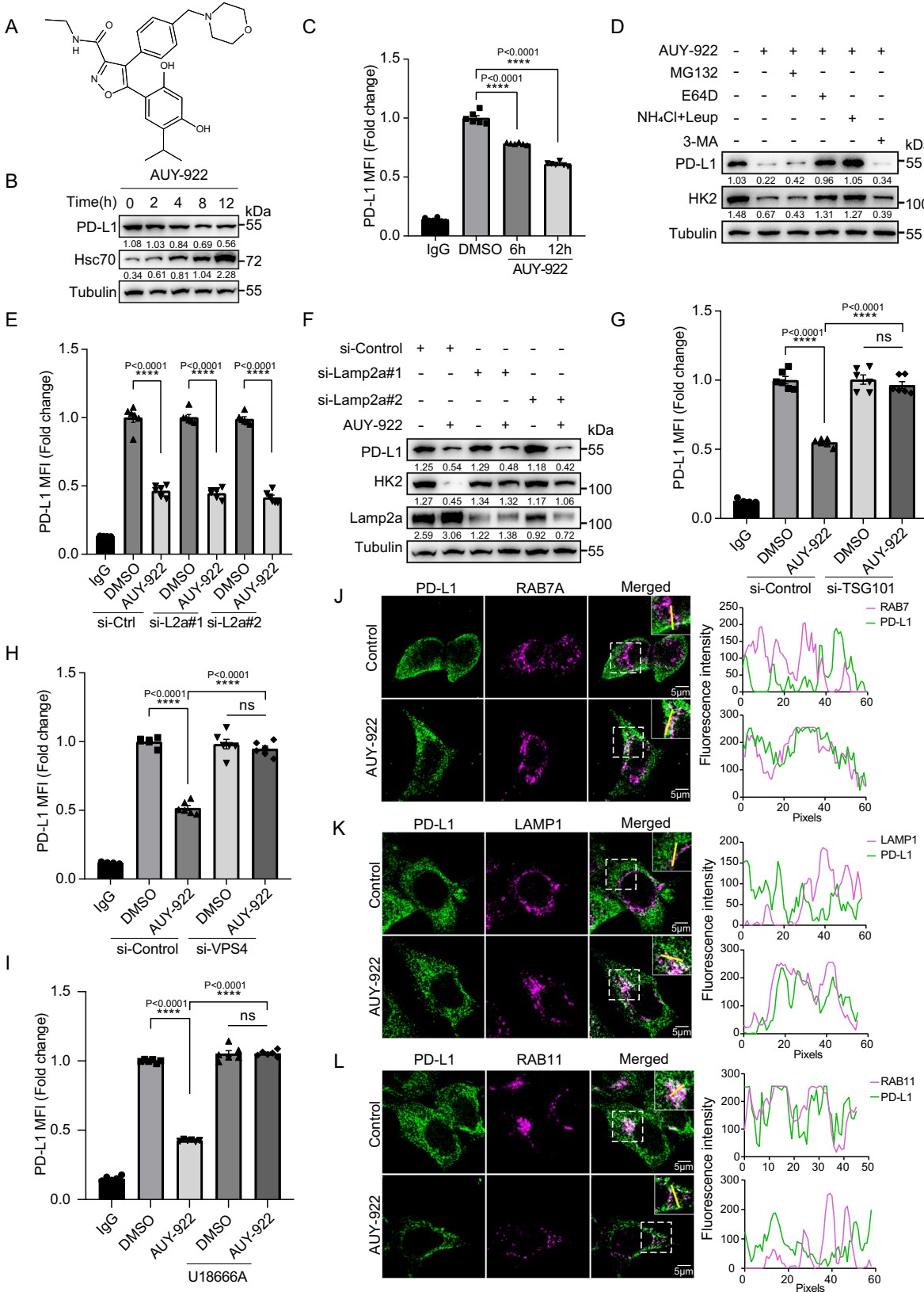

secondary antibodies for immunofluorescence were used: goat anti-rabbit Alexa Fluor 546 (#A-11035, 1:500, Thermo Fisher Scientific), goat anti-mouse Alexa Fluor 488 (#A-11029, 1:500, Thermo Fisher Scientific), goat anti-mouse DyLight 649 (#A23610, 1:500, Abbkine). The in vivo antibodies for mouse models were used: control antibody InVivoMab rat IgG2b isotype (#BE0090; 100 μg; Bioxcell), anti-PD-L1 Ab (#BE0101; 100 μg; Bioxcell), anti-CTLA4 Ab (#BE0131; 75 μg; Bioxcell).

## Mass spectrometry

HEK293T cells were transfected with Flag, PD-L1-Flag (pcDNA5 with CMV promoter) or Hsc70-WT, Hsc70-3KA (pcDNA5 with CMV

**Fig. 6 | AUY-922 promotes PD-L1 degradation through Hsc70-mediated eMI.**
**A** Structural formula of AUY-922. **B** MCF-7 cells were treated with 1 μM AUY-922 for
0, 2, 4, 8, and 12 h. PD-L1 levels were detected by western blotting. **C** MCF-7 cells
were treated with 1 μM AUY-922 for 6 h and 12 h, fluorescence of PD-L1 on the cell
membrane was analyzed by flow cytometry, $n = 6$. **D** MCF-7 cells were treated with
1 μM AUY-922 for 12 h, treated with or without MG132 (10 μM), $NH_4Cl$ (20 mM),
Leupeptin (100 nM), 3-MA (5 mM), E-64D (10 μM) for another 12 h. Cell lysates were
detected by western blotting. **E, F** MCF-7 cells were transfected with siRNAs of
Lamp2a for 48 h, treated with or without 1 μM AUY-922 for another 12 h. Fluores-
cence of PD-L1 on the cell membrane was analyzed by flow cytometry, $n = 6$ (**E**), cell
lysates were detected by western blotting (**F**). **G** MCF-7 cells were transfected with
siRNAs of TSG101 for 48 h, treated with or without 1 μM AUY-922 for another 12 h.
Fluorescence of PD-L1 on the cell membrane was analyzed by flow cytometry, $n = 6$.
**H** MCF-7 cells were transfected with siRNAs of VPS4 for 48 h, treated with or
without 1 μM AUY-922 for another 12 h. Fluorescence of PD-L1 on the cell

membrane was analyzed by flow cytometry, $n = 6$. **I** MCF-7 cells were treated with
1 μM AUY-922 with or without 5 μg/mL U18666A for 12 h. Fluorescence of PD-L1 on
the surface of cell membrane was analyzed by flow cytometry, $n = 6$. **J–L** MCF-7 cells
were treated with 1 μM AUY-922 for 4 h and the co-localization between RAB7A and
PD-L1 (**J**), LAMP1 and PD-L1 (**K**), RAB11 and PD-L1 (**L**) were done using immuno-
fluorescence and confocal microscopy. Scale bar, 5 μm. The intensity profiles along
the yellow line are plotted in the middle panels, with the colocalizing sites marked
by white. For (**C, E, G–I**), MCF-7 cells were seeded in 48-well plate with 6 replicates
per group and subjected to the corresponding treatment, repeated independently
three times and similar results were obtained. Data shown in (**B, D, F**) and (**J–L**) were
repeated independently three times with similar results. Data represent Mean ±
SEM, for (**C, E, G–I**), $n = 6$, two-sided with adjustment of Tukey's multiple com-
parisons, one-way ANOVA, $P$ value is indicated in the graph. The numbers under
blots represent the value (the ratio to Tubulin) of grayscale quantification. Source
data are provided as a Source data file.

promoter) for 24 h. Cells were washed with PBS and lysed in 1 mL lysis
buffer (TAP) (20 mM Tris-HCl (pH 7.5), 150 mM NaCl, 0.5% NP-40, 1 mM
NaF, 1 mM $Na_3VO_4$, 1 mM EDTA, protease inhibitor cocktail (Bimake,
add fresh) for 30 min, incubated with anti-Flag (DYKDDDDK) beads for
6 h on a rotating wheel at 4 °C. The beads were washed with TAP three
times, 5 min each wash. Flag-tagged PD-L1 or Flag-tagged Hsc70 were
trypsin-digested and protein identification and quantitation was
through LC-MS/MS.

For details, the immunoprecipitated proteins were resolved in 8 M
urea and 500 mM Tris-HCl (pH 8.5). Disulfide bridges were reduced by
adding Tris (2-carboxyethyl) phosphine (TCEP) at a final concentration
of 5 mM for 20 min. Reduced cysteine residues were then alkylated by
adding 10 mM iodoacetamide (IAA) and incubating for 15 min in the
dark at room temperature. The urea concentration was reduced to 2 M
by adding 100 mM Tris-HCl (pH 8.5) and 1 mM $CaCl_2$. The protein
mixture was digested overnight at 37 °C with trypsin (V5111, Promega) at
an enzyme-to-substrate ratio of 1:100 (w/w). The resulting peptides was
analyzed on a Thermo Scientific Q Exactive HF-X mass spectrometer.

The protein identification and quantification were done by
MaxQuant v1.5.8[47]. For details, the tandem mass spectra were sear-
ched against the UniProt human protein database (downloaded in
Mar 2020, 20, 286 entries) and the built-in contaminant protein list.
Trypsin was set as the enzyme, and the specificity was set to both N
and C terminal of the peptides. The maximum missed cleavage was
set to 2. The cysteine carbamidomethylation was set as a static
modification, and the methionine oxidation was set as a variable
modification. The precursor and fragment mass tolerance were set as
20 ppm. The first-search peptide mass tolerance and main-search
peptide tolerance were set to 20 and 4.5 ppm, respectively. The false
discovery rate at the peptide spectrum match level and protein level
was controlled to be <1%. The "match between runs" option and iBAQ
module were applied. Only unique peptides and razor peptides were
used for quantification, and the minimum ratio count for protein
identification was 2. The summed peptide intensities were used for
protein quantification.

### Molecular cloning and siRNA knockdown
Hsc70-Flag/HA (WT and 3KA), Lamp2a-Flag, TFG-Flag and PD-L1-Flag/
HA (WT and mutants) were cloned into pcDNA5 via BamHI and AvaI
restriction enzymes by PCR amplifying the ORFs from cDNA tem-
plates of Hsc70 (#19514, Addgene), Lamp2a (#86146, Addgene), TFG
(#CH877169, WZ Biosciences Inc) and PD-L1 (#121466, Addgene).
Molecular cloning was performed using T4 DNA ligase and trans-
formed into DH5α E. coli cells. Plasmid purifications and extractions
were performed using the NucleoBond Xtra Midi kit (Macherey-
Nagel). siRNA was transformed into HEK293T, MCF-7, PANC1 cells
with Lipofectamine™ 3000 (Invitrogen), according to the manu-
facturer's protocol. The sequences of siRNA are provided in Sup-
plementary Table 1.

### qRT-PCR analysis
Cells were lysed and total RNA was extracted using TRIzol™ Plus RNA
Purification Kit (Invitrogen). cDNA was synthesized from purified RNA
using the PrimeScript RT reagent Kit (TAKARA, RR047A) according to
the manufacturer's instructions. Quantitative PCR was performed
using a StepOnePlus Rela-Time PCR Systems (ABI). The comparative Ct
method was used for the data analysis and human Tubulin mRNA was
used as an internal control. The sequences of primers used for qRT-
PCR are provided in Supplementary Table 2.

### Flow cytometry analysis of membrane PD-L1
For flow cytometry analysis of membrane PD-L1, 4T1, U937, or MCF-7
cells were seeded in 48-well plate with at least six replicates per
group and subjected to the corresponding treatment, and then col-
lected by centrifugation at $1000 \times g$ for 5 min, incubated with PBS
(0.5% BSA) for 10 min at room temperature. The cells were probed
with PE anti-mouse CD274 (MIH6) (#153612, 1:200, Biolegend) or PE
anti-human CD274 (29E.2A3) (#329706, Biolegend) and a matched
isotype at 4 °C for 30 min in the dark. After washing three times with
PBS, the cells were analyzed using flow cytometry (Beckman Coulter
Cytoflex), and data were analyzed using software CytExpert 2.4 and
FlowJo v10.8.1.

### Immunoprecipitation
pcDNA5-PD-L1-WT-Flag, pcDNA5-PD-L1#1-Flag, pcDNA5-PD-L1#2-Flag,
pcDNA5-PD-L1#3-Flag, pcDNA5-PD-L1#4-Flag, pcDNA5-PD-L1#5-Flag,
pcDNA5-PD-L1#6-Flag, pcDNA5-CMTM6-Flag or pcDNA5-TFG-Flag, the
indicated plasmids were transfected into HEK293T cells for 24 h. Cells
were washed with PBS and lysed in 1 mL lysis buffer (TAP) (20 mM Tris-
HCl (pH 7.5), 150 mM NaCl, 0.5% NP-40, 1 mM NaF, 1 mM $Na_3VO_4$, 1 mM
EDTA, protease inhibitor cocktail (Bimake, add fresh) for 30 min,
incubated with anti-Flag (DYKDDDDK) beads for 6 h on a rotating
wheel at 4 °C. The beads were washed with TAP three times, 5 min each
wash, and SDS-PAGE 2X loading buffer was added, followed by heating
at 100 °C for 10 min.

### Isolation of membrane proteins and cytoplasmic proteins
MCF-7 cells were pretreated with IFN-γ for 48 h, followed by trans-
fection with Flag, Hsc70-WT-Flag or Hsc70-3KA-Flag for additional
24 h. Alternatively, cells were treated with AUY-922 for another 12 h.
Subsequently, cells were collected and the extraction of cytoplasmic
and membrane proteins were performed according to the instructions
provided by Membrane and Cytoplasmic Protein Extraction kit
(C510005, Sangon Biotech, China).

### Immunoblotting
Cell lysates or pull-down samples were added SDS-PAGE 2X loading
buffer and heated at 100°C for 10 min, subjected to 10–12% SDS-
PAGE, and then transferred onto PVDF membranes for 1 h at 0.2 A

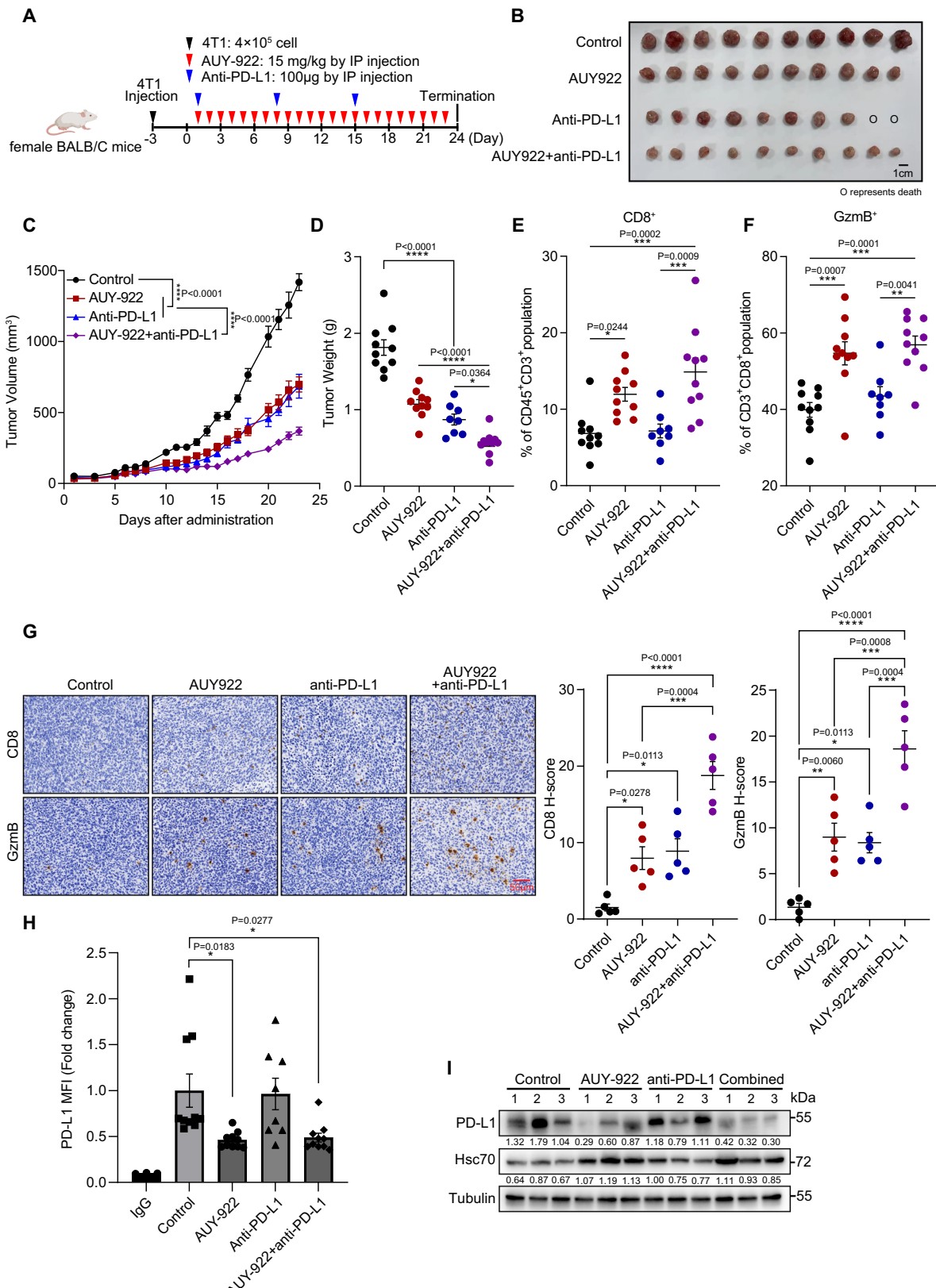

O represents death

with semi-dry transfer system of BioRad. Membranes were blocked in PBST buffer containing 10% (w/v) skimmed milk for 1 h and probed with the indicated antibodies in PBST containing 5% (w/v) BSA at 4 °C overnight. Detection was performed using HRP-conjugated secondary antibodies and chemiluminescence reagents (#4 AW001-500, 4A Biotech Co.). We used ImageJ 1.53a for quantitative analysis of grayscale. The numbers under the blots represent the value (the ratio to Tubulin).

### Immunofluorescence

For immunofluorescent assays, the time points when PD-L1 had not undergone significant degradation were selected. These time points

**Fig. 7 | AUY-922 promotes anti-tumor immunity and enhances the therapeutic effect of aPD-L1 ICI. A** BALB/c mice were inoculated with $4 \times 10^5$ 4T1 cells, group administration according to the time points shown in the schematic diagram, created with BioRender.com, released under a Creative Commons Attribution-NonCommercial-NoDerivs 4.0 International license. **B–D** Tumor growth of Control ($n = 10$), AUY-922 (15 mg/kg, $n = 10$), Anti-PD-L1 (100 μg, $n = 10$) and the combination of AUY-922 and anti-PD-L1(AUY-922+anti-PD-L1, $n = 10$) in BALB/c mice and final tumor weights. (O represents death at the third dose of Anti-PD-L1 administration). **E, F** Flow cytometry analysis of CD8$^+$ T cells (**E**) and CD8$^+$GzmB$^+$ T cells (**F**) in tumors, Control ($n = 10$), AUY-922 ($n = 10$), Anti-PD-L1 ($n = 8$), AUY-922+anti-PD-L1 ($n = 10$). **G** Immunohistochemistry analysis of CD8 and GzmB in the 4T1 tumors.

Scale bar, 50 μm. $n = 5$ (each group has 5 tumor tissues with the average of 3 random fields). **H** Flow cytometry analysis of PD-L1 levels on the surface of cell membrane in tumor cells, Control ($n = 10$), AUY-922 ($n = 10$), Anti-PD-L1 ($n = 8$), AUY-922+anti-PD-L1 ($n = 10$). **I** Indicated protein levels were detected by Immunoblotting with the indicated harvested tumor cells. $n = 3$ (randomly selected tumor tissues from 3 mice in each group for detection) and repeated independently two times with similar results. Data represent Mean ± SEM, for (**C**) two-way ANOVA, for (**D–H**) one-way ANOVA, two-sided with adjustment of Tukey's multiple comparisons, $P$ value is indicated in the graph. The numbers under blots represent the value (the ratio to Tubulin) of grayscale quantification. Source data are provided as a Source data file.

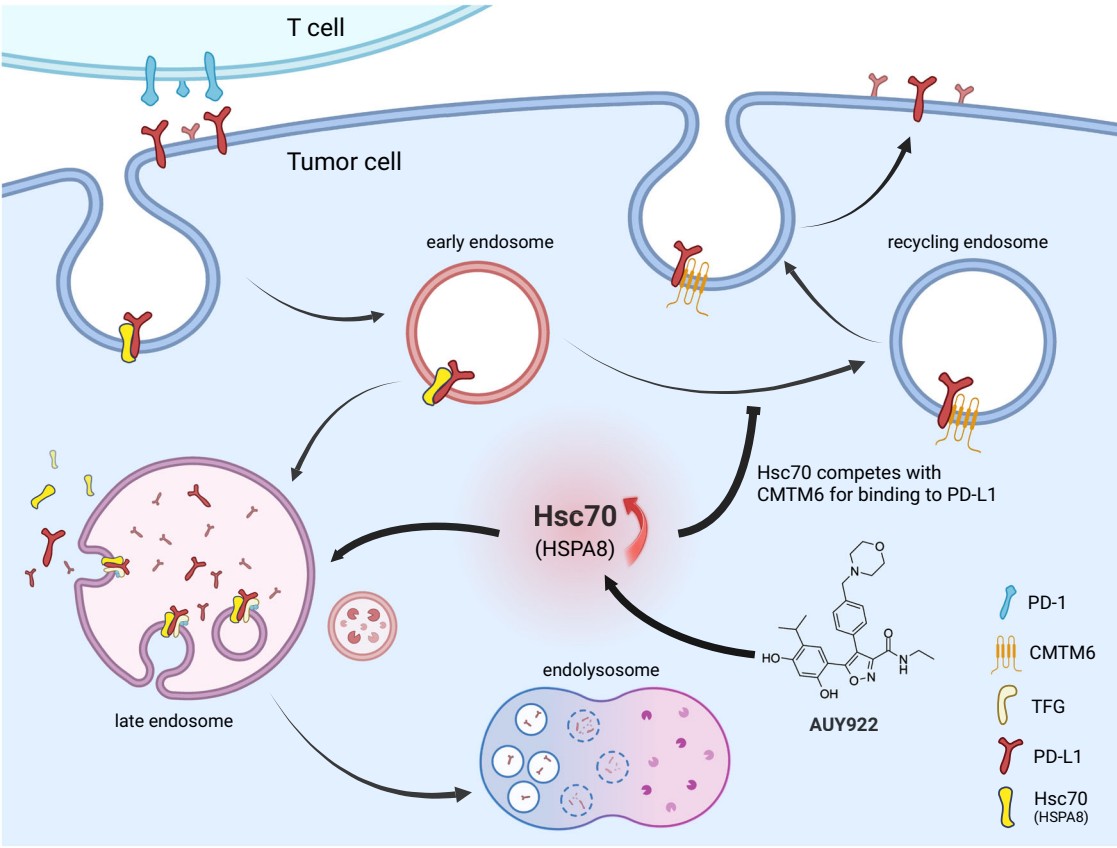

**Fig. 8 | Schematic model of Hsc70 promoted anti-tumor immunity by targeting PD-L1 degradation through eMI.** Endogenous-level Hsc70 plays a critical role in mediating basal levels of PD-L1 recycling and degradation. An elevated expression of Hsc70 or an augmentation in Hsc70 levels induced by AUY-922 disrupts the fundamental PD-L1 recycling mechanism, results in enhanced internalization of cytoplasmic PD-L1 by Hsc70, leading to its transport to late endosomes where it undergoes degradation by lysosomes. Furthermore, Hsc70 competes with CMTM6 for binding sites on PD-L1, thus diminishing PD-L1 to recycle back to the cell membrane via CMTM6. Together, this diminished recycling reduces the interaction between PD-L1 on tumor cells and PD-1 on T cells, ultimately reactivating T cell-mediated anti-tumor immunity. Graph was created with BioRender.com, released under a Creative Commons Attribution-NonCommercial-NoDerivs 4.0 International license.

are conductive to observing the co-localization of PD-L1 with specific organelles during its degradation process. MCF-7 cells were cultured on glass slides, fixed with 4% paraformaldehyde for 20 min and permeabilized using PBS containing 0.1% Triton X-100 for 1 h at room temperature. Cells were washed three times with PBS, and blocked with 5% BSA in PBS for 1 h at room temperature. Cells were incubated with anti-LAMP1 (1:100), anti-RAB11 (1:50) or anti-RAB7A (1:100) and anti-PD-L1 (1:100) antibodies overnight at 4 °C. After washing three times with PBS, the cells were incubated with goat anti-rabbit Alexa Fluor 546 (#A-11035, 1:500, Thermo Fisher Scientific) and goat anti-mouse Alexa Fluor 488 (#A-11029, 1:500, Thermo Fisher Scientific) for 1 h. Images were obtained using FV3000 OLYMPUS microscope.

## Generation of stable 4T1 cell lines which overexpress Hsc70-WT (Hsc70-WT-OE), Hsc70-3KA (Hsc70-3KA-OE), knockdown PD-L1 with or without Hsc70 overexpression

Briefly, HEK293T cells were transfected with pCDH-CTRL, pCDH-Hsc70-WT, pCDH-Hsc70-3KA packaging plasmids for overexpression of Hsc70. In addition, HEK293T cells were transfected with pLent-sh-CTRL, pLent-sh-PD-L1 packaging plasmids for knockdown of PD-L1. The supernatant was collected at 48, 72, and 96 h, filtered with 0.45 μm filters and precipitate the virus with PEG8000. 4T1 cells were infected with virus and the medium was replaced after 24 h post-infection. The infected 4T1 cells were selected with 4 μg/mL puromycin and separated into single cells by flow cytometry.

## Ethical approval statement

All mice were housed in a specific pathogen-free (SPF) facility and were euthanized with carbon dioxide. All the animal experiments were strictly conducted in accordance with the protocols approved by the Tab of Animal Experimental Ethical Inspection of the First Affiliated Hospital, College of Medicine, Zhejiang University. The tumor size allowed in the experiment did not exceed 2000 mm$^3$, we adhere to this limit to ensure that the animals did not suffer undue harm and that the experimental results are valid and reproducible.

## In vivo tumor models and treatment

Female BALB/c mice or nude mice (aged 8–10 weeks) were housed in suitable temperature and humidity environment (25 °C, suitable humidity (typically 50%), 12 h dark/light cycle), and fed with sufficient water and food.

For tumor models with overexpression of Hsc70 or knockdown of PD-L1 with or without overexpression Hsc70, $4 \times 10^5$ indicated 4T1 stable cells were suspended in 100 μL PBS and Matrigel (1:1 v/v) and injected into the 2th breast fat pad. On day 3, tumor size was measured and calculated by using the formula: 1/2 × length × width$^2$. Tumor weight was recorded on the day of sacrifice.

For tumor models with AUY-922 with or without anti-PD-L1 Ab and anti-CTLA4 Ab treatment, $4 \times 10^5$ 4T1 cells were suspended in 100 μL PBS and Matrigel (1:1 v/v) and injected into the 2th breast fat pad. When the tumor volume reaches 75 mm$^3$, the mice with similar tumor burdens were randomized into treatment groups. AUY-922 (20/15 mg/kg, i.p.) was dissolved in menstruum (5% DMSO + 30% PEG400 + 65% ddH$_2$O), and given daily after grouped; anti-PD-L1 Ab (100 μg, i.p.) and anti-CTLA4 Ab (75 μg, i.p.) were administered every 7 days after grouped. Tumor size was measured and calculated by using the formula: 1/2 × length × width$^2$. Tumor weight was measured at the end of the experiment.

## Tumor sample preparation, flow cytometry, and immunoblotting

Tumors were collected and processed into single-cell suspensions through digestion in collagenase type I (#2350118, Gibco) and DNase I (#143582, Roche) at 37 °C for 45 min. After filtering with a 45 μm filter (BD Bioscience), the isolated cells were stained with the specific surface marker antibodies, Zombie Violet™ Fixable Viability Kit (#423114; Biolegend), anti-CD45-PerCP-Cy5.5 (#103132; Biolegend), anti-CD3-PE-Cy7(#100320; Biolegend) and anti-CD8-FITC (#100706; Biolegend) in PBS for 30 min at 4 °C. Intracellular staining of GzmB was performed as follows: cells were washed and then fixed and permeabilized with a Fix/Perm kit (#421403; Biolegend), and finally stained with anti-APC-GzmB (#372204; Biolegend). For proper compensation of flow cytometry channels, single-stain samples were utilized. For immunoblotting, after digesting tumor tissues with collagenase type I and DNase I, and filtering the resulting cells through 45 μm filter, 100 μL of total cells suspension is extracted. An equal volume of 2×loading buffer is then added, and the mixture is boiled at 100 °C for 10 min to prepare the total tumor lysates.

## Immunohistochemistry

Tumor tissues were rapidly excised, fixed with 4% paraformaldehyde, and embedded in paraffin for tissue sections and immunohistochemical staining. The primary antibodies used are anti-CD8 and anti-GzmB. Visualization of cell nuclei was performed with hematoxylin and analysis was done using the Olympus BX61 light microscope. The statistical analysis was performed by image J.

## Statistics and reproducibility

Statistical analyses were performed with GraphPad Prism 8.0. Data and analyzed with one-way ANOVA (where more than two groups of data were compared) or *t*-test (where two groups of data were compared) and *P*-values were indicated in the related graphs. For the tumor volume in animal experiments, a two-way ANOVA analysis was used to determine statistical significance. Data points are shown as Mean ± SEM. All experiments were repeated independently at least two or three times and similar results were obtained.

## Reporting summary

Further information on research design is available in the Nature Portfolio Reporting Summary linked to this article.

## Data availability

The mass spectrometry proteomics data generated in this study have been deposited in the ProteomeXchange Consortium via the iProX partner repository with the dataset identifier PXD051281 https://www.iprox.cn//page/project.html?id=IPX0008562000 and PXD051241 https://www.iprox.cn//page/project.html?id=IPX0008528000. The remaining data are available within the Article, Supplementary Information or Source data file. Source data are provided with this paper.

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

## Acknowledgements

Financial support from the National Natural Science Foundation of China (32222023, 32350016 and 82272855 to H.X., 32100598 to X.C.), the Natural Science Foundation of Zhejiang Province (LR22C070002 to H.X., LQ24C070001 to X.X. and LR22H160011 to J.Z.), the Key Research and Development Program of Zhejiang (2024SSYS0020 and 2022C03005 to H.X.), Leading Innovative and Entrepreneur Team Introduction Program of Zhejiang (2021R01012 to H.X., 2023R01012 to D.W.), Leading innovation and entrepreneurship team of Hangzhou (TD2020006 to H.X.).

## Author contributions

H.X. conceived and coordinated the project and designed the experiments. D.W. and H.X. interpreted the data and wrote the manuscript. X.X., T.X., M.Z., Y.S. and F.W. performed most of the experiments. Y.T., Z.C., Y.X., R.W., X.C., J.Z., T.H., L.Z., C.H. and Q.Z. assisted with the experiments and helped to analyze the data.

## Competing interests

X.X., H.X., T.X., M.Z., Y.X. and Z.C. have filed a patent application "the use of drug combinations, 202311744204.1". The remaining authors declare no other competing interests.
