## [Peer Review File · Nature Communications]

Hsc70 promotes anti-tumor immunity by targeting PD-L1 for lysosomal degradationREVIEWER COMMENTS

Reviewer #1 (Remarks to the Author): with expertise in cancer immunology, PD-L1 regulation

In this manuscript, the authors demonstrated that Hsc70 (official symbol: HSPA8), a member of the heat shock protein 70 family, promotes PD-L1 degradation via the endosomal-lysosomal pathway. Additionally, the authors found that a small molecule HSP90 inhibitor, AUY922, induces Hsc70 expression, reduces PD-L1 expression, and further improves the efficacy of PD-L1 blockade treatment in a mouse tumor model. Overall, the study convincingly demonstrates the role of Hsc70 in the regulation of PD-L1 protein levels. However, there are some overclaim and concerns that need to be addressed.

1. The statement in the title, "Hsc70 promotes anti-tumor immunity by targeting PD-L1 for lysosomal degradation," is not supported by the current set of data. The authors have shown that Hsc70 overexpression reduces tumor growth in immune competent mice but did not provide evidence to support that Hsc70 functions through PD-L1. As the authors mentioned in the introduction, Hsc70 has many substrates. To evaluate whether the functional consequence of Hsc70 overexpression is specifically through PD-L1, two additional groups should be included in Figure 5B: Group 1, "Hsc70-WT overexpression + PD-L1 knockout," and Group 2, "PD-L1 knockout."

2. Figure 1A indicates that many housekeeping genes interact with PD-L1, which is interesting and surprising. However, CMTM6, a known interacting partner of PD-L1, did not appear on the list. A detailed method used for the experiment should be provided, which is currently lacking in the manuscript. Additionally, a control experiment using PD-L1 knockout cells should be performed.

3. There is no evidence of a reduction in PD-L1 in Hsc70 WT overexpression cells compared to Flag control cells in Figure 2I, 2J, 2K, 6I, 6J, nor any reduction of PD-L1 in CMTM6 knockdown cells in Figure 3J, 3K, and 3L. However, Western blot data show a clear reduction in both cases. Could the authors explain this discrepancy?

4. PD-L1 is a membrane protein, but most of the immunofluorescence images show intracellular staining only. A PD-L1 knockdown or knockout control should be provided to demonstrate the specificity of the anti-PD-L1 antibody used here. Furthermore, the authors should state the number of replicate experiments performed in total in the figure legends and whether the data shown in the figures are representative of all replicates for all immunofluorescence experiments.

5. In Figure 3C, there is a weak band detected by the Hsc70 antibody in the IP condition of Flag "+" and si-CMTM6 "+". This is unexpected. Could the authors explain this? Additionally, information regarding the control Flag construct should be provided.

6. Supplementary Figure 3F and G: Could the authors explain the multiple bands in the lanes - PD-L1-WT, PD-L1#3-6?

7. Supplementary Figure 1A and Figure 7A: Is there any evidence that the antibody used for blocking does not compete for the binding of the detection antibody? The methods used for these experiments should be described in more detail.

Reviewer #2 (Remarks to the Author): with expertise in cancer immunology, PD-L1 regulation

General Comments

In this study, the authors identified HSC70 as a novel regulator of PD-L1 expression. They suggested that HSC70 competes with CMTM6, disrupting the surface recycling of PD-L1, leading to its lysosomal degradation through endoplasmic reticulum microautophagy (eMI). Through animal models, the authors showed that either the overexpression of HSC70 or the induction of its expression following treatment with the small molecule AUY-922 can effectively degrade PD-L1 in tumor tissues, thereby activating antitumor immunity. These findings hint at a potential therapeutic strategy that combines targeting HSC70 with immune checkpoint blockers. These findings can be considered novel and has the potential to expand the possibilities within cancer immunotherapy. However, in its current form, the manuscript might not be ready for direct publication in esteemed journals like Nature

Communications. To assist the authors, I have provided some comments to improve the clarity and robustness of their hypothesis, which I hope will help strengthen the scientific validity of the evidence presented.

Major comments

1. Highly expressed housekeeping genes or chaperone proteins are generally more likely to be captured by bulk IDs, so caution is warranted when considering chaperones as target proteins. The authors' results in Fig. 1A could be a similar case as well, but the rationale for selecting HSC70 from several top candidates is not detailed in the text. This needs to be explained in more detail to strengthen the justification for the research target.
2. The authors demonstrate in supplementary Figure 1C that the expression of PD-L1 mRNA is not affected according to HSC70 level, and they focus on the regulation of PD-L1 at the post-translation protein level. If they can show the effect of HSC70 on exogenous PD-L1 overexpressed by promoters other than the endogenous PD-L1 promoter, such as CMV, it would help to clearly determine that the effect of HSC70 on PD-L1 is not at the transcriptional level but due to post-translational modifications (PTM)
3. The authors have identified the degradation of PD-L1 by HSC70 through the endosome-lysosome pathway. Therefore, they need to carefully edit the imaging data to clearly show the location changes of PD-L1 in these organelles. They should re-edit primarily with magnified images where one can clearly see both the organelles and PD-L1 in Figure 2 I, J, K / 3 J, K, L / 4 I, J and 6 J, K, L
4. If the expression of PD-L1 is reduced due to the KD of CMTM6 and HSC70 in the imaging data of Figure 3 J, K, L / 4 I, J, and 6 J, K, L, it would have been difficult to observe the location of PD-L1. PD-L1 does not appear to be reduced compared to Cont si. It is necessary to check whether the experiment was conducted in a state where the degradation of PD-L1 was blocked by treating with an inhibitor. If information is missing, describe the information in the legend.
5. In previous studies, HSP90 has been shown to play a crucial role in the transcription of PD-L1. Furthermore, HSP90 inhibitors are known to reduce the expression of PD-L1 at the transcriptional level. As AUY-922 was developed as an HSP90 inhibitor, it is essential to explore the likelihood that it regulates PD-L1 transcriptionally. However, the authors have demonstrated that AUY-922 induces eMI degradation of PD-L1 via HSC70 expression, not

just its HSP90 inhibition. This was further substantiated by findings in supplementary Fig4 E, F, which showed the effect of AUY-922 being entirely neutralized when HSC-70 underwent knockdown (KD). If AUY-922's impact on PD-L1 primarily arises from HSC70, sidelining mRNA level regulation, a more in-depth discussion on the potential mechanism by which AUY-922's HSP90 inhibition heightens HSC70 expression, and reasons why this inhibition doesn't influence PD-L1 transcription, could greatly enhance the paper's quality.

6. It seems reasonable to think that instead of combining AUY-922 with anti-PD-L1, it would be more effective to pair it with blockers of non-PD-1/PD-L1 signals like anti-CTLA4 and anti-TIM3. Such a combination is expected to produce a better synergistic effect. It would be beneficial to have supporting data on the efficacy when AUY-922 is combined with other immune checkpoint inhibitors (ICBs).

7. For the readers' understanding, add a model summarizing the elucidated mechanism of action.

Minor comments

1. In the introduction, providing information about the functions and differences between CAM and eMI would help readers understand better.

2. In Supplementary Fig. 1A, a masking effect could occur if the treated antibody and the antibody used in the FACS analysis recognize the same region of PD-L1. Information on whether the two antibodies recognize different epitopes of PD-L1 should be provided. Alternatively, performing live imaging using PD-L1 antibodies labeled with different fluorescent dyes in combination with a lysotracker may provide an answer.

3. Please pay attention to the labels so that one can intuitively understand the conditions and what is being measured by looking at the picture. . ex) F1-f: Lamp2a => Lamp2a-frag, F1-j, SF3-D_Y-axis: MFI => PD-L1 MFI

4. In Fig. 2E and 3A, the band signal looks saturated, so the change in PD-L1 is not clear. It is needed to replace them with the same picture at a lower exposure.

Reviewer #3 (Remarks to the Author): with expertise in heat shock proteins

This manuscript describes a very interesting study on the fate of PD-L1 that is targeted as a strategy to ameliorate cancer that has demonstrated great success. The therapeutic

intervention is aimed at down-regulating the presence of PD-L1 from the surface of cancer cells. However, this therapeutic approach has some challenges due to the development of resistance due, in part, to the recycling of PD-L1 to the cell surface after internalization, allowing the degradation of the protein. Therefore, investigations directed at enhancing PD-L1 degradation are of great importance. Indeed, this study was aimed to understand and develop approaches to degrade PD-L1. Consequently, the study is of high relevance.

The authors found that Hsc70 (HSPA8), a constitutive member of the heat shock family of proteins, is associated with PD-L1 and could direct the receptor to the lysosome for degradation. The study is well performed using a variety of controls and techniques to validate their initial hypothesis. Thus, the investigation is solid and mechanistic. The main point is that HSPA8 competes with CMTM6, a natural partner of PD-L1, for binding to PD-L1. Due to the capacity of HSPA8 to recognize phosphatidylserine phospholipids (PS) present on the cytosolic side of cellular membranes, it may help in directing PD-L1 to the lysosome for breaking down.

Although the study is well performed, there are several aspects that should be addressed. If the authors decide to revise the manuscript, it is very likely that its impact will be enhanced.

Specific comments:

1. Among all proteins identified by the mass spectrographic analysis (Fig. 1A), several other chaperones were present, including HSPA1 and HSP90. HSPA1 is abundant in cancer cells and has an overlapping function with HSPA8, particularly in its ability to bind to PS. The authors should explain the rationale for selecting HSPA8 over HSPA1.

2. HSPA8 is one of the most abundant proteins in the cytosol, with a concentration of 107 molecules per cell. Could the authors explain why it is necessary to overexpress the protein for the effect on PD-L1? Are the constitutive levels not enough for targeting PD-L1?

3. How much do the HSPA8 levels increase overexpression compared to basal levels? A simple Western blot may do the job.

4. The authors used a commercial polyclonal antibody to HSPA8. Are the authors sure that there is no cross-reaction with HSPA1? Maybe they should demonstrate the lack of cross-reaction using commercially available recombinant proteins for HSPA8 and HSPA1.

5. The reviewer is concerned that overexpression of HSPA8 may be toxic for the cell, as it has been reported in the past (see Feder et al. 1992, *Genes Dev* 6:1402–1413; Arispe et al. 2004, *FASEB J* 18:1636–1645). Did the authors check cellular viability after overexpression of HSPA8?

6. The authors concluded that HSPA8 and CMTM6 bind to the same side on PD-L1, within aa 161-241. Would the authors indicate that the region is in the PD-L1, cytosolic, or membrane domain?

7. Mechanistically, the authors assume that HSPA8 is binding to PD-L1, perhaps at the cytosolic side (not mentioned in the manuscript), and target the endocytic vesicle to the lysosome, avoiding receptors recycling. If the reviewer got this concept right, could the authors explain or speculate the mechanism involved? Is HSPA8 interacting with a specific component of lysosomes? The authors may like to check the manuscript by Kirkegaard et al. 2010, *Nature* 463:549–553, and Mahalka et al. 2014, *Biochim Biophys Acta* 1838:1344–1361, which showed the interaction of HSPA1 with the negatively charged phospholipid BMP, which is a major component of the lysosome membrane. The authors may also like to check an excellent review of the interaction of HSP with membranes (De Maio and Hightower 2021, *Cell Stress Chaperones* 26:769-783).

8. The reviewer is concerned that the observations from this study on the involvement of HSPA8 in vesicular traffic are solely due to this protein, or perhaps HSPA1 is also playing a role. HSPA1 is very abundant in cancer cells but not in normal tissues. Therefore, it may be a more appropriate target.

9. The authors have shown that the HSP90 inhibitor AUY-922 increases the expression of HSPA8. However, it is well established that these inhibitors elevate the expression of HSPA1 by increasing the transcriptional activity of HSF1. Moreover, increasing the expression of

HSPA1 by stress or by HSP90 inhibitors resulted in an increase in the endocytic/vesicular process (see Vega et al. 2010 Cell Stress Chaperones 15:517–527). There are not prior reports, to the best of the reviewer, that HSPA8 expression increases upon inhibition of HSP90. These observations may indeed point to the direction of a possible role of HSPA1.

10. The manuscript will be greatly enhanced if a model (even speculative) of the process of HSPA8 interaction with PD-L1 and targeting the lysosome is presented. It may be a great resource for experts and non-experts on the subject.

Reviewer #4 (Remarks to the Author): with expertise in heat shock proteins, immunology

Paper titled "Hsc70 promotes anti-tumor immunity by targeting PD-L1 for lysosomal degradation" identifies new molecular mechanism of Hsc70-mediated PDL-1 lysosomal degradation and provides new approach to improve efficacy of checkpoint inhibition therapy (PD-1/PD-L1). As pointed by the authors, just very recently was uncovered that after aPD-L1 treatment, PD-L1 is re-expressed on the surface of the cell and this mechanism confers resistance to ICI therapy. Hence, controlling the re-expression of PD-L1 is very important and this paper provides new avenues for reducing the amount of PD-L1 recirculating back to the cell membrane. This work presents very original findings that Hsp90 inhibitor, AUY922 facilitates PD-L1 degradation via lysosomes and boost anti-tumor immune responses by augmenting Hsc70- expression.

Methodology used is appropriate and provides enough details for this work to be reproduced.

I do not have any major, just a few minor concerns.

1. Authors should include summary graph for all data showing representative surface expression of PD-L1 by flow cytometry (Fig 1C and H; Fig 2B and D, F and H; Fig 3B and Fig 7A)
2. Authors should state that reported MFI values on y-axes' are PD-L1 MFI throughout the manuscript (Fig 1F and J; Fig 3 E, G and I; Fig 4 H; Fig 6 E, G, H and I; Fig 7 H).
3. In Fig 7 F and H there is no additive effect on the frequency of CD8+ T cells or Gzb+ CD8+ T cells after combination of AUY-922 + anti-PD-L1 (there is no statistical significance

between AUY-922 and AUY-922+anti-PD-L1). How do you explain this (since there is statistically significant effect on the tumor volume and weight). Also, in H, there is no significant difference on the surface expression levels of Pd-L1 between AUY and AUY+anti-PD-L1.

4. Legend of Fig 5: CD8 is not a CTL marker, should be replaced with surface marker

Overall, I recommend this paper for publication.

REVIEWER COMMENTS

Reviewer #1 (Remarks to the Author): with expertise in cancer immunology, PD-L1 regulation

In this manuscript, the authors demonstrated that Hsc70 (official symbol: HSPA8), a member of the heat shock protein 70 family, promotes PD-L1 degradation via the endosomal-lysosomal pathway. Additionally, the authors found that a small molecule HSP90 inhibitor, AUY922, induces Hsc70 expression, reduces PD-L1 expression, and further improves the efficacy of PD-L1 blockade treatment in a mouse tumor model. Overall, the study convincingly demonstrates the role of Hsc70 in the regulation of PD-L1 protein levels. However, there are some overclaim and concerns that need to be addressed.

Response:

We thank the reviewer for the appreciation of our study and the constructive comments. We have addressed the comments in detail below, we have also performed new experiments, added 46 new experimental panels to the manuscript and additional 18 new experiment panels in Response Letter. The text, newly added to the manuscript, is indicated in red. We hope that the reviewer finds our response satisfactory. Please note that the figure citations in our responses below refer to the new (post-revision) figures.

1. The statement in the title, "Hsc70 promotes anti-tumor immunity by targeting PD-L1 for lysosomal degradation," is not supported by the current set of data. The authors have shown that Hsc70 overexpression reduces tumor growth in immune competent mice but did not provide evidence to support that Hsc70 functions through PD-L1. As the authors mentioned in the introduction, Hsc70 has many substrates. To evaluate whether the functional consequence of Hsc70 overexpression is specifically through PD-L1, two additional groups should be included in Figure 5B: Group 1, "Hsc70-WT overexpression + PD-L1 knockout," and Group 2, "PD-L1 knockout."

Response:

We thank the reviewer for the constructive comments.

As suggested, we examined the individual and combined effects of PD-L1 knockdown and Hsc70 overexpression on tumor growth. We established stable transgenic 4T1 cell lines that knocked down PD-L1 with or without overexpression of wild-type Hsc70, and inoculated them into immunocompetent BALB/c mice (Supplementary Fig 5A). Our findings revealed that Hsc70 overexpression (Hsc70-OE) or PD-L1 knockdown (PD-L1-KD) effectively suppressed tumor growth, while the combination of both (Hsc70-OE+PD-L1-KD) did not further enhance the inhibitory effect (Supplementary Fig 5B-D). These results suggest that Hsc70 overexpression may not have an additive effect on PD-L1 knockdown. Our findings also demonstrated a marked increase in the levels of CD8⁺ and activated CD8⁺ cells in tumors with PD-L1 knockdown or Hsc70 overexpression (Supplementary Fig 5E-G). PD-L1 levels in the harvested tumor cells showed a dramatic decrease after Hsc70 overexpression (Supplementary Fig 5H). Collectively, these findings suggest that the

function of Hsc70 overexpression on anti-tumor T cell response is mainly mediated through PD-L1.

We have provided these information in the results (Supplementary Fig. 5) and methods of revised manuscript.

The corresponding results are shown as follows:

Supplementary Fig 5 The anti-tumor consequence of Hsc70 overexpression is mainly through PD-L1

(A) The schematic diagram of 4T1 breast cancer tumor model.

(B-D) Tumor growth of Control (n=10), PD-L1-KD (n=10), Hsc70-OE (n=10) and Hsc70-OE+PD-L1-KD (n=10) group which injected with indicated cells in BALB/c mice and final tumor weights.

(E, F) Flow cytometry analysis of CD8⁺ T cells (E) and CD8⁺GzmB⁺ T cells (F) in tumors.

(G) Immunohistochemistry analysis of CD8 and GzmB in the 4T1 tumors. Scale bar, 50 μ m. n=15 (each group has 5 tumor tissues, and each tissue contain 3 random fields).

(H) Indicated protein levels were detected by Immunoblotting with the indicated harvested tumor cells.

Data represent Mean \pm SEM. *p<0.05, **p<0.01, ****p<0.0001.

The numbers under blots represent the value (the ratio to Tubulin) of grayscale quantification.

2. Figure 1A indicates that many housekeeping genes interact with PD-L1, which is interesting and surprising. However, CMTM6, a known interacting partner of PD-L1, did not appear on the list. A detailed method used for the experiment should be provided, which is currently lacking in the manuscript. Additionally, a control experiment using PD-L1 knockout cells should be performed.

Response:

Thank you for your valuable feedback.

The IP-mass spectrometry was performed in HEK293T cells transfected with Flag or PD-L1-Flag (CMV promoter). IP was performed using anti-Flag beads, and Flag-transduced HEK293T cells were used as the control. CMTM6 is a membrane protein (Burr, Sparbier et al. 2017, Mezzadra, Sun et al. 2017), has a relatively low expression in total cell proteome. Our mass spectrometry approach was designed to evaluate the entire proteome rather than membrane proteome-specific, which will explain why CMTM6 was not detected in our mass spectrometry data.

The detailed method used for the experiment was added in the methods of revised manuscript.

3. There is no evidence of a reduction in PD-L1 in Hsc70 WT overexpression cells compared to Flag control cells in Figure 2I, 2J, 2K, 6I, 6J, nor any reduction of PD-L1 in CMTM6 knockdown cells in Figure 3J, 3K, and 3L. However, Western blot data show a clear reduction in both cases. Could the authors explain this discrepancy?

Response:

Thank you for the comments.

We apologized for any confusion caused by some incomplete description in the figure legends. To clarify, for PD-L1 detection in the immunofluorescence experiment, we selected the time points before the degradation of PD-L1; in contrast, evaluation of PD-L1 in western blotting analyses were performed at later time points.

We have now processed western blotting under the same time points as the immunofluorescence experiment. The results confirmed that PD-L1 levels had not yet

undergone degradation at the selected time points (Response Letter Fig 1-3). We appreciate your feedback which helps to improve the clarity and accuracy of our manuscript. We have now included the information in the figure legends on the western blotting and immune fluorescence assays.

The corresponding results are shown as follows:

Response Letter Fig 1 PD-L1 levels remained unchanged after overexpression of Hsc70 for 8 h

MCF-7 cells were transfected with Flag, Hsc70-WT-Flag or Hsc70-3KA-Flag for 8 h, cell lysates were immunoblotted with indicated antibodies.

Response Letter Fig 2 PD-L1 levels remained unchanged after knocking down CMTM6 for 36 h

MCF-7 cells were transfected with siRNA of Hsc70 or CMTM6 for 36 h, cell lysates were immunoblotted with indicated antibodies.

Response Letter Fig 3 PD-L1 levels remained unchanged after 4 h of treatment with AUJ-922

MCF-7 cells were treated with 1 μ M AUJ-922 for 4 h, cell lysates were immunoblotted with indicated antibodies.

4. PD-L1 is a membrane protein, but most of the immunofluorescence images show intracellular staining only. A PD-L1 knockdown or knockout control should be provided to demonstrate the specificity of the anti-PD-L1 antibody used here. Furthermore, the authors should state the number of replicate experiments performed in total in the figure legends and whether the data shown in the figures are representative of all replicates for all immunofluorescence experiments.

Response:

We are grateful to the reviewer for their valuable feedback. We have now included a PD-L1 knockdown control to demonstrate the specificity of the anti-PD-L1 antibody with immunofluorescence. As shown in the Response Letter Fig 4, IFN- γ stimulation significantly enhances PD-L1 expression, primarily localized on the cell membrane. This observation aligns with previous reports that IFN- γ stimulation increased the levels of PD-L1 on the cell membrane (Kim, Myers et al. 2005, Knopf, Stowbur et al. 2023). Meanwhile, upon knocking down PD-L1 under IFN- γ stimulation, we observed a sharp decrease in the fluorescence level of PD-L1, further confirming the specificity of the PD-L1 antibody.

Regarding our immunofluorescence data, we have at least three replicates for all experiments. Furthermore, as shown in Supplementary Fig 2I-K; 3I-K; 4A, B; 6O-Q, we have analyzed the Pearson coefficients representing interactions using three replicates. These results were added in the Supplementary Figures and legends of revised manuscript.

The corresponding results are shown as follows:

Response Letter Fig 4 IFN- γ stimulation increases the surface expression of PD-L1, while knocking down PD-L1 significantly reduces the fluorescence intensity of PD-L1

MCF-7 cells were pretreated with IFN- γ for 48 h, then transfected with siRNA of PD-L1 for 48 h, perform immunofluorescence staining on PD-L1 and take photos with FV3000.

Supplementary Fig 2I-K; 3I-K; 4A, B; 6O-Q The Pearson coefficients representing interactions were analyzed by three replicates of immunofluorescence data

Supplementary Fig 2I-K

Supplementary Fig 3I-K

Supplementary Fig 4A, B

Supplementary Fig 6O-Q

Supplementary Fig 2I-K MCF-7 cells were transfected with Flag, Hsc70-WT-Flag or Hsc70-3KA-Flag for 8 h, the co-localization between RAB7A and PD-L1 (I), Lamp1 and PD-L1 (J), RAB11 and PD-L1 (K) were analyzed by Pearson correlation coefficient with three replicates.

Supplementary Fig 3I-K MCF-7 cells were transfected with indicated siRNA for 36 h and the co-localization between RAB7A and PD-L1(I), LAMP1 and PD-L1 (J), RAB11 and PD-L1 (K) were analyzed by Pearson correlation coefficient with three replicates.

Supplementary Fig 4A-B MCF7 were transfected with siRNA of TFG for 48 h, then transfected with Hsc70 for another 8 h, the co-localization between RAB7A and PD-L1 (A), LAMP1 and PD-L1 (B) were analyzed by Pearson correlation coefficient with three replicates.

Supplementary Fig 6O-Q MCF-7 cells were treated with 1 μ M AUY-922 for 4 h and the co-localization between RAB7A and PD-L1 (O), the co-localization between LAMP1 and PD-L1 (P), the co-localization between RAB11 and PD-L1 (Q) were analyzed by Pearson correlation coefficient with three replicates.

5. In Figure 3C, there is a weak band detected by the Hsc70 antibody in the IP condition of Flag "+" and si-CMTM6 "+". This is unexpected. Could the authors explain this? Additionally, information regarding the control Flag construct should be provided.

Response:

Thank you for pointing this detail out. The faintly visible band detected by the Hsc70 antibody might have resulted from non-specific binding due to overexposure. To address this issue, we have repeated the entire experiment and carefully adjusted the exposure conditions. The revised **Figure 3C** clearly demonstrates the specific binding of the antibody, without any non-specific signals.

Regarding the construction of the Flag or PD-L1-Flag plasmids, these were prepared with the pcDNA5 vector, which employs the CMV promoter for gene expression. This vector enables high-level expression of the target gene. We have added this information in **Methods**.

The corresponding results are shown as follows:

Figure 3C Knockdown CMTM6 enhances the interaction of Hsc70 and PD-L1

HEK293T cells were transfected with siRNA of CMTM6 for 48 h, then transfected with Flag or PD-L1-Flag in another 12 h, the interaction between Hsc70 and PD-L1 was detected by immunoprecipitation.

6. Supplementary Figure 3F and G: Could the authors explain the multiple bands in the lanes - PD-L1-WT, PD-L1#3-6?

Response:

Thank you for your comments.

PD-L1 undergoes various modifications within cells, including polyubiquitination and glycosylation (Shi, Wang et al. 2022). Previous studies have shown that PD-L1 exhibit non-glycosylated (MW-33kDa) and glycosylated PD-L1 (MW-50kDa) (Chan, Li et al. 2019, Shi, Wang et al. 2022, Li, Lim et al. 2016, Wang, Lee et al. 2020). Consistent with these findings, our results also show the presence of glycosylated and non-glycosylated bands for PD-L1-WT (Response Letter Fig 5).

PD-L1 contains multiple glycosylation sites, including N35, N192, N200, N219. Our results showed that N35 may be the main glycosylation site of PD-L1. PD-L1 #1-2 lacking N35 site excision only showed a single band. PD-L1 #3-6 containing N35 site displays multiple bands which may represent different levels of glycosylation. It is worth noting that the other glycosylation sites of PD-L1 will more or less affect the glycosylation of PD-L1. This may be a rather complex mechanism that requires more in-depth research.

The corresponding results are shown as follows:

Response Letter Fig 5 PD-L1 exhibits different modifications of glycosylation and non-glycosylation

HKE293T cells were transfected with Hsc70-HA and PD-L1-Flag (CMV promoter) for 24 h, cell lysates were immunoblotted with indicated antibodies. Black circle, glycosylated PD-L1; arrowhead, non-glycosylated PD-L1.

7. Supplementary Figure 1A and Figure 7A: Is there any evidence that the antibody used for blocking does not compete for the binding of the detection antibody? The methods used for these experiments should be described in more detail.

Response: We appreciate the reviewer's valuable comments.

In our experiment, the blocking antibody was obtained from Bioxcell with clone number 10F.9G2™, while the detection antibody was sourced from Biolegend with clone number MIH6. We have revised this information in Methods of revised manuscript.

To further investigate whether these two clones recognize the same epitope, we incubated IFN- γ -treated 4T1 cells with the anti-PD-L1 Ab (10F.9G2™) for varying durations, and the surface fluorescence intensity of PD-L1 was analyzed by flow cytometry using antibodies of the same clone (10F.9G2) or the MIH6 clone. As shown in **Response Letter Fig 6**, when using the 10F.9G2 clone for detection, PD-L1 intensity drops to the same level as IgG control within 1 min. In contrast, when the MIH6 clone was used for detection, the fluorescence intensity of PD-L1 gradually decreased, with detectable value after 1-min of pre-incubation and baseline level after 5 min. The dynamics of PD-L1 intensity detected using the MIH6 clone is consistent with the rapid internalization of PD-L1 after antibody pre-incubation which showed that PD-L1 internalization occurred within 400 s (Li, Lim et al. 2018), suggesting that the binding of MIH6 clones was unlikely to be interfered by the 10F.9G2 clone.

The corresponding results are shown as follows:

Response Letter Fig 6 the antibody 10F.9G2 used for blocking does not compete for the binding of the detection antibody MIH6

4T1 cells were pretreated with 100 ng/ml IFN- γ for 48 h, and subsequently incubated with anti-PD-L1 Ab (12.5 μ g/ml) for varying durations of 0, 1, 5, 15 and 30 min, collect cells after cleaning with PBS, flow cytometry staining using two PD-L1 monoclonal antibodies (10F.9G2, MIH6) conjugated with PE.

Reference:

Burr, M. L., C. E. Sparbier, Y. C. Chan, J. C. Williamson, K. Woods, P. A. Beavis, E. Y. N. Lam, M. A. Henderson, C. C. Bell, S. Stolzenburg, O. Gilan, S. Bloor, T. Noori, D. W. Morgens, M. C. Bassik, P. J. Neeson, A. Behren, P. K. Darcy, S. J. Dawson, I. Voskoboinik, J. A. Trapani, J. Cebon, P. J. Lehner and M. A. Dawson (2017). "CMTM6 maintains the expression of PD-L1 and regulates anti-tumour immunity." *Nature* **549**(7670): 101-105.

Chan, L. C., C. W. Li, W. Xia, J. M. Hsu, H. H. Lee, J. H. Cha, H. L. Wang, W. H. Yang, E. Y. Yen, W. C. Chang, Z. Zha, S. O. Lim, Y. J. Lai, C. Liu, J. Liu, Q. Dong, Y. Yang, L. Sun, Y. Wei, L. Nie, J. L. Hsu, H. Li, Q. Ye, M. M. Hassan, H. M. Amin, A. O. Kaseb, X. Lin, S. C. Wang and M. C. Hung (2019). "IL-6/JAK1 pathway drives PD-L1 Y112 phosphorylation to promote cancer immune evasion." *J Clin Invest* **129**(8): 3324-3338.

Kim, J., A. C. Myers, L. Chen, D. M. Pardoll, Q. A. Truong-Tran, A. P. Lane, J. F. McDyer, L. Fortunato and R. P. Schleimer (2005). "Constitutive and inducible expression of b7 family of ligands by human airway epithelial cells." *Am J Respir Cell Mol Biol* **33**(3): 280-289.

Kornepati, A. V. R., R. K. Vadlamudi and T. J. Curiel (2022). "Programmed death ligand 1 signals in cancer cells." *Nat Rev Cancer* **22**(3): 174-189.

Knopf, P., D. Stowbur, S. H. L. Hoffmann, N. Hermann, A. Maurer, V. Bucher, M. Poxleitner, B. Tako, D. Sonanini, B. Krishnamachary, S. Sinharay, B. Fehrenbacher, I. Gonzalez-Menendez, F. Reckmann, D. Bomze, L. Flatz, D. Kramer, M. Schaller, S. Forchhammer, Z. M. Bhujwala, L. Quintanilla-Martinez, K. Schulze-Osthoff, M. D. Pagel, M. F. Fransen, M. Rocken, A. F. Martins, B. J. Pichler, K. Ghoreschi and M. Kneilling (2023). "Acidosis-mediated increase in IFN-gamma-induced PD-L1 expression on cancer cells as an immune escape mechanism in solid tumors." *Mol Cancer* **22**(1): 207.

- Li, C. W., S. O. Lim, E. M. Chung, Y. S. Kim, A. H. Park, J. Yao, J. H. Cha, W. Xia, L. C. Chan, T. Kim, S. S. Chang, H. H. Lee, C. K. Chou, Y. L. Liu, H. C. Yeh, E. P. Perillo, A. K. Dunn, C. W. Kuo, K. H. Khoo, J. L. Hsu, Y. Wu, J. M. Hsu, H. Yamaguchi, T. H. Huang, A. A. Sahin, G. N. Hortobagyi, S. S. Yoo and M. C. Hung (2018). "Eradication of Triple-Negative Breast Cancer Cells by Targeting Glycosylated PD-L1." Cancer Cell **33**(2): 187-201 e110.
- Li, C. W., S. O. Lim, W. Xia, H. H. Lee, L. C. Chan, C. W. Kuo, K. H. Khoo, S. S. Chang, J. H. Cha, T. Kim, J. L. Hsu, Y. Wu, J. M. Hsu, H. Yamaguchi, Q. Ding, Y. Wang, J. Yao, C. C. Lee, H. J. Wu, A. A. Sahin, J. P. Allison, D. Yu, G. N. Hortobagyi and M. C. Hung (2016). "Glycosylation and stabilization of programmed death ligand-1 suppresses T-cell activity." Nat Commun **7**: 12632.
- Mezzadra, R., C. Sun, L. T. Jae, R. Gomez-Eerland, E. de Vries, W. Wu, M. E. W. Logtenberg, M. Slagter, E. A. Rozeman, I. Hofland, A. Broeks, H. M. Horlings, L. F. A. Wessels, C. U. Blank, Y. Xiao, A. J. R. Heck, J. Borst, T. R. Brummelkamp and T. N. M. Schumacher (2017). "Identification of CMTM6 and CMTM4 as PD-L1 protein regulators." Nature **549**(7670): 106-110.
- Shi, C., Y. Wang, M. Wu, Y. Chen, F. Liu, Z. Shen, Y. Wang, S. Xie, Y. Shen, L. Sang, Z. Zhang, Z. Gao, L. Yang, L. Qu, Z. Yang, X. He, Y. Guo, C. Pan, J. Che, H. Ju, J. Liu, Z. Cai, Q. Yan, L. Yu, L. Wang, X. Dong, P. Xu, J. Shao, Y. Liu, X. Li, W. Wang, R. Zhou, T. Zhou and A. Lin (2022). "Promoting anti-tumor immunity by targeting TMUB1 to modulate PD-L1 polyubiquitination and glycosylation." Nat Commun **13**(1): 6951.
- Wang, Y. N., H. H. Lee, J. L. Hsu, D. Yu and M. C. Hung (2020). "The impact of PD-L1 N-linked glycosylation on cancer therapy and clinical diagnosis." J Biomed Sci **27**(1): 77.

Reviewer #2 (Remarks to the Author): with expertise in cancer immunology, PD-L1 regulation

General Comments

In this study, the authors identified HSC70 as a novel regulator of PD-L1 expression. They suggested that HSC70 competes with CMTM6, disrupting the surface recycling of PD-L1, leading to its lysosomal degradation through endoplasmic reticulum microautophagy (eMI). Through animal models, the authors showed that either the overexpression of HSC70 or the induction of its expression following treatment with the small molecule AUY-922 can effectively degrade PD-L1 in tumor tissues, thereby activating antitumor immunity. These findings hint at a potential therapeutic strategy that combines targeting HSC70 with immune checkpoint blockers. These findings can be considered novel and has the potential to expand the possibilities within cancer immunotherapy. However, in its current form, the manuscript might not be ready for direct publication in esteemed journals like Nature Communications. To assist the authors, I have provided some comments to improve the clarity and robustness of their hypothesis, which I hope will help strengthen the scientific validity of the evidence presented.

Response:

We thank the reviewer for the appreciation of our study and the constructive comments. We have addressed the comments in detail below, we have also performed new experiments, added 46 new experimental panels to the manuscript and additional 18 new experiment panels in Response Letter. The text, newly added to the manuscript, is indicated in red. We hope that the reviewer finds our response satisfactory. Please note that the figure citations in our responses below refer to the new (post-revision) figures.

Major comments

1. Highly expressed housekeeping genes or chaperone proteins are generally more likely to be captured by bulk IDs, so caution is warranted when considering chaperones as target proteins. The authors' results in Fig. 1A could be a similar case as well, but the rationale for selecting HSC70 from several top candidates is not detailed in the text. This needs to be explained in more detail to strengthen the justification for the research target.

Response:

We express our gratitude to the reviewer for their valuable feedback. We chose Hsc70 mainly based on the following considerations:

Firstly, we agree with the reviewer that some housekeeping genes like α -tubulin, β -tubulin or actin may show non-specific binding to PD-L1 due to their abundant expression in cell. Therefore, we excluded these proteins from the list.

Secondly, we used Kaplan-Meier Plotter to analyze several top-ranked proteins in PD-L1 mass spectrometry data, including HSPA1, EEEF1A1, FUS, HSPA90AA1, HSPA90AB1 and HSPA8. As shown in **Supplementary Fig 1B**, among these proteins, only the high

expression of HSPA8 is positively correlated with the overall survival rate of tumor patients.

Taking into account all the aforementioned considerations, we have chosen HSPA8 (also known as Hsc70) as the focal point for our research endeavors. We have now revised the description of how this gene was selected in the results and Supplementary Fig 1B of revised manuscript.

The corresponding results are shown as follows:

Supplementary Fig 1B only HSPA8 expression is positively correlated with OS in all tumor types of patients who have not received ICIs

Correlation analysis between HSPA8, EEF1A1, FUS, HSPA1, HSP90AA1, HSP90AB1 gene levels with overall survival (OS) in all tumor types of patients who have not received any immune checkpoint inhibitors (ICIs) treatment from the Kaplan-Meier Plotter platform ([Kaplan-Meier plotter \[Immunotherapy\] \(kmplot.com\)](https://kmplot.com/)).

2. The authors demonstrate in supplementary Figure 1C that the expression of PD-L1 mRNA is not affected according to HSC70 level, and they focus on the regulation of PD-L1 at the post-translation protein level. If they can show the effect of HSC70 on exogenous PD-L1 overexpressed by promoters other than the endogenous PD-L1 promoter, such as CMV, it would help to clearly determine that the effect of HSC70 on PD-L1 is not at the transcriptional level but due to post-translational modifications (PTM)

Response:

We thank the reviewer for the constructive comments.

As shown in Supplementary Fig 1F, we utilized HEK293T cells (with low endogenous PD-L1 expression) and overexpressed PD-L1 using a CMV promoter-driven plasmid (pcDNA5). In this setting, an additional overexpression of Hsc70 decreased exogenous PD-L1 levels (tagged with Flag), indicating a post-translational regulation. We have now added the corresponding results in Supplementary Fig 1F for better clarity.

The corresponding results are shown as follows:

Supplementary Fig 1F overexpression of Hsc70 promoted the degradation of exogenous PD-L1

HEK293T cells were transfected with indicated plasmids for 24 h, cell lysates were immunoblotted with indicated antibodies.

3. The authors have identified the degradation of PD-L1 by HSC70 through the endosome-lysosome pathway. Therefore, they need to carefully edit the imaging data to clearly show the location changes of PD-L1 in these organelles. They should re-edit primarily with magnified images where one can clearly see both the organelles and PD-L1 in Figure 2 I, J, K / 3 J, K, L / 4 I, J and 6 J, K, L

Response: thank you for the comments.

we have re-edited the immunofluorescence experiment images to provide a magnified view as shown in Figure 2I-K; Figure 3J-L; Figure 4I-J; Figure 6J-L.

Meanwhile, we acknowledge the limitations of fluorescence microscopy in resolving the fine structural details of organelles. For instance, the diameter of lysosomes ranges from 0.025 to 0.8 μm , mitochondria measure approximately 1-2 μm in diameter, and late endosomes have an average diameter of 0.3 μm (Mullock, Bright et al. 1998). To overcome these limitations, we employed additional assays to further support our conclusions:

- i) We characterized the corresponding organelles using specific biomarkers. For instance, RAB7A serves as a biomarker for late endosomes, RAB11 for recycling endosomes, and LAMP1 for lysosomes, aiming to provide a more accurate representation of the subcellular localization of PD-L1 within the immune fluorescence images.
- ii) As shown in Figure 1D, only lysosome inhibitor can restore PD-L1 expression in

the context of Hsc70 overexpression.

- iii) As shown in Figure 2A-H, treatment of MCF-7 cells with U18666A or knockdown of TSG101 and VPS4 resulted in the blockade of PD-L1 degradation.

These findings consistently implicate the endosome-lysosome pathway in the degradation of PD-L1 mediated by Hsc70, which we believe could overcome the limitations of immune fluorescence staining and together provide adequate support for our statements.

The corresponding results are as follows:

Figure 2I-K overexpression of Hsc70 increased the co-localization of PD-L1 with late endosomes and lysosomes, decreased the co-localization of PD-L1 with recycling endosomes

MCF-7 cells were transfected with Flag, Hsc70-WT-Flag or Hsc70-3KA-Flag for 8 h, the co-localization between RAB7A and PD-L1 (I), the co-localization between LAMP1 and PD-L1 (J), the co-localization between RAB11 and PD-L1 (K) were done using immunofluorescence and confocal microscopy. Scale bar, 5 μm. The intensity profiles along the yellow line are plotted in the middle panels, with the colocalizing sites marked by white.

Figure 3J-L knockdown of CMTM6 increased the co-localization of PD-L1 with late endosomes and lysosomes, decreased the co-localization of PD-L1 with recycling endosomes. However, knockdown of Hsc70 reversed it

MCF-7 cells were transfected with indicated siRNA for 36 h and the co-localization between RAB7A and PD-L1 (J), the co-localization between LAMP1 and PD-L1 (K), the co-localization between RAB11 and PD-L1 (L) were done using immunofluorescence and confocal microscopy. Scale bar, 5 μm. The intensity profiles along the yellow line are plotted in the middle panels, with the colocalizing sites marked by white.

Figure 4 I-J knockdown of TFG blocked the increased co-localization of PD-L1 with late endosomes and lysosomes under overexpression of Hsc70

MCF-7 cells were transfected with siRNA of TFG for 48 h, then transfected with Hsc70 for another 8 h, the co-localization between RAB7A and PD-L1 (I), the co-localization between LAMP1 and PD-L1 (J) were analyzed by immunofluorescence and confocal microscopy. Scale bar, 5 μm. The intensity profiles along the yellow line are plotted in the middle panels, with the colocalizing sites marked by white.

Figure 6 J-L AUY-922 treatment increased co-localization of PD-L1 with late endosomes and lysosomes, decreased the co-localization of PD-L1 with recycling endosomes

MCF-7 cells were treated with 1 μ M AUY-922 for 4 h and the co-localization between RAB7A and PD-L1 (J), the co-localization between LAMP1 and PD-L1 (K), the co-localization between RAB11 and PD-L1 (L) were done using immunofluorescence and confocal microscopy. Scale bar, 5 μ m. The intensity profiles along the yellow line are plotted in the middle panels, with the colocalizing sites marked by white.

4. If the expression of PD-L1 is reduced due to the KD of CMTM6 and HSC70 in the imaging data of Figure 3 J, K, L / 4 I, J, and 6 J, K, L, it would have been difficult to observe the location of PD-L1. PD-L1 does not appear to be reduced compared to Cont si. It is necessary to check whether the experiment was conducted in a state where the degradation of PD-L1 was blocked by treating with an inhibitor. If information is missing, describe the information in the legend.

Response:

Thank you for pointing out the potential misunderstanding in our figure legends. We apologize for any confusion this may have caused. To clarify, in the experiments described, we did not employ inhibitors to prevent the degradation of PD-L1. Instead, for PD-L1 detection in the immunofluorescence experiment, we selected the time points before the degradation of PD-L1; in contrast, evaluation of PD-L1 in western blotting analyses were performed at later time points.

We have now processed western blotting under the same time points as the immunofluorescence experiment. The results confirmed that PD-L1 levels had not yet undergone degradation at the selected time points (Response Letter Fig 1-3, Figure 4F, G). We appreciate your feedback which helps to improve the clarity and accuracy of our manuscript. We have now included the information in the figure legends on the western blotting and immune fluorescence assays.

The corresponding results are shown as follows:

Response Letter Fig 1 PD-L1 levels remained unchanged after overexpression of Hsc70 for 8 h

MCF-7 cells were transfected with Flag, Hsc70-WT-Flag or Hsc70-3KA-Flag for 8 h, cell lysates were immunoblotted with indicated antibodies.

Response Letter Fig 2 PD-L1 level was not changed when knocking down CTMT6

or Hsc70 for 36 h

MCF-7 cells were transfected with indicated siRNA for 36 h, cell lysates were immunoblotted with indicated antibodies.

Figure 4 F, G Knocking down TFG prevents Hsc70 mediated PD-L1 degradation

(F) MCF-7 cells were transfected with siRNA of TFG for 60 h, indicated protein levels were detected by western blotting.

(G) MCF7 were transfected with siRNA of TFG for 36 h, then transfected with Hsc70 for another 24 h, indicated protein levels were detected by western blotting.

Response Letter Fig 3 PD-L1 levels remained unchanged after 4 h of treatment with AUY-922

MCF-7 cells were treated with 1 μM AUY-922 for 4 h, cell lysates were immunoblotted with indicated antibodies.

5. In previous studies, HSP90 has been shown to play a crucial role in the transcription of PD-L1. Furthermore, HSP90 inhibitors are known to reduce the expression of PD-L1 at the transcriptional level. As AUY-922 was developed as an HSP90 inhibitor, it is essential to explore the likelihood that it regulates PD-L1 transcriptionally. However, the authors have demonstrated that AUY-922 induces eMI degradation of PD-L1 via HSC70 expression, not just its HSP90 inhibition. This was further substantiated by findings in supplementary Fig4 E, F, which showed the effect of AUY-922 being entirely neutralized when HSC-70

underwent knockdown (KD). If AUY-922's impact on PD-L1 primarily arises from HSC70, sidelining mRNA level regulation, a more in-depth discussion on the potential mechanism by which AUY-922's HSP90 inhibition heightens HSC70 expression, and reasons why this inhibition doesn't influence PD-L1 transcription, could greatly enhance the paper's quality.

Response:

Thank you for your valuable feedback. To investigate whether HSP90 inhibitors regulate PD-L1 transcriptionally, we treated MCF-7, A375 and Hela cells with HSP90 inhibitors AUY-922 and 17-AAG for 12 hours, and evaluated the mRNA levels of Hsc70 and PD-L1 by qPCR. To ensure data accuracy, we utilized two distinct primer sets for PD-L1. We found that treatment with AUY-922 or 17-AAG led to a decrease of PD-L1 mRNA in IFN- γ -treated cells, but an increase of PD-L1 mRNA in untreated cells. These findings suggest that the effect of HSP90 inhibitors on PD-L1 transcription might be dependent on IFN- γ -mediated PD-L1 transcriptional activity. Furthermore, since Hsc70 is able to downregulate PD-L1 protein in either IFN- γ -treated or -untreated cells (Figure 6B, C; Supplementary Fig 6A-D), we would infer that the regulation occurred at post-translational rather than transcriptional level. The results are shown in Response Letter Fig 7.

To further confirm that the degradation of PD-L1 occurs at the post-translational level, we treated MCF-7 cells with cycloheximide (CHX), a protein synthesis inhibitor, to block the translation of PD-L1. Subsequently, we exposed the cells to AUY-922 treatment. As shown in Response Letter Fig 8, AUY-922 accelerated the degradation of PD-L1, indicating that its effect on PD-L1 degradation occurs independently of transcriptional regulation.

To further strengthen this observation, we transfected MCF-7 cells with a pCMV-3-PD-L1-Flag plasmid containing a CMV promoter for 24 h. Following this, the cells were treated with 1 μ M AUY-922 for an additional 12 h. Western blotting (WB) analysis revealed that AUY-922 concurrently degraded PD-L1 levels (Response Letter Fig 9), confirming its role in post-translational modifications.

Furthermore, our results also confirmed that AUY-922 promotes PD-L1 degradation via Hsc70-mediated endosomal microautophagy (eMI), as evident from Figure 6 and Supplementary Fig 6. Collectively, these findings suggest that the degradation of PD-L1 by AUY-922 is attributed to post-translational modifications.

The corresponding results are shown as follows:

Response Letter Fig 7 HSP90 inhibitor (AUY-922/17-AAG) treatment decrease mRNA levels of PD-L1 in IFN-treated cells, but increase mRNA levels of PD-L1 in untreated cells
--

MCF-7, A375 and HeLa cells were pretreated with or without IFN- γ (100 ng/mL) for 48 h, treated with 17-AAG (1 μ M) or AUY-922 (1 μ M) for another 12 h, mRNA levels of PD-L1 and Hsc70 were analyzed by qPCR.

Response Letter Fig 8 AUY-922 treatment accelerated the degradation of PD-L1

MCF-7 cells were treated with CHX for 6 h and 12 h, and treated with or without AUY-922 for 12 h, cell lysates were immunoblotted by indicated antibodies.

Response Letter Fig 9 AUY-922 treatment degraded exogenous expression of PD-L1

MCF-7 cells were transfected with indicated plasmids for 24 h, then treated with AUY-922 (1 μ M) for 12 h, cell lysates were immunoblotted with indicated antibodies.

6. It seems reasonable to think that instead of combining AUY-922 with anti-PD-L1, it would be more effective to pair it with blockers of non-PD-1/PD-L1 signals like anti-CTLA4 and anti-TIM3. Such a combination is expected to produce a better synergistic effect. It would be beneficial to have supporting data on the efficacy when AUY-922 is combined with other immune checkpoint inhibitors (ICBs).

Response:

Thank you for your valuable feedback. We have now performed anti-CTLA4 treatment in the presence/absence of AUY-922 (Supplementary Fig 8A). Reduction of tumor growth was observed in either AUY-922 or anti-CTLA4 monotherapy, but a further decrease of tumor growth was observed in the AUY-922 and anti-CTLA4 combinational treatment group (Supplementary Fig 8B-D). The levels of total and activated intratumoral CD8+ cytotoxic T cells were significantly increased upon combinational treatment compared to anti-CTLA4 monotherapy (Supplementary Fig 8E-G). Combinational treatment or AUY-922 alone reduced the surface expression of PD-L1 and upregulated Hsc70 levels (Supplementary Fig 8H-I). Collectively, our results have suggested a potential drug candidate for combinational immunotherapy with ICI treatment.

The corresponding results are shown as follows:

Supplementary Fig 8 AUY-922 promotes antitumor immunity and enhances the therapeutic effect of anti-CTLA4

(A) BALB/c mice were inoculated with 4×10^5 4T1 cells, group administration according to the time points shown in the schematic diagram.

(B-D) Tumor growth of Control, AUY922 (15 mg/kg), Anti-CTLA4 (75 μ g) and the combination of AUY-922 and anti-CTLA4 (AUY-922+anti-CTLA4) in BALB/c mice and final tumor weights.

(E, F) Flow cytometry analysis of CD8⁺ T cells (E) and CD8⁺GzmB⁺ T cells (F) in tumors.

(G) Immunohistochemistry analysis of CD8 and GzmB in tumors. Scale bar, 50 μ m. n=15 (each group has 5 tumor tissues, and each tissue contain 3 random fields).

(H) Flow cytometry analysis of PD-L1 levels on the surface of cell membrane in tumor cells.

(l) Indicated protein levels were detected by Immunoblotting with the indicated harvested tumor cells.

Data represent Mean \pm SEM. **p<0.01, ***p<0.001, ****p<0.0001.

The numbers under blots represent the value (the ratio to Tubulin) of grayscale quantification.

7. For the readers' understanding, add a model summarizing the elucidated mechanism of action.

Response:

Thank you for the comments. We have now added **Figure 8** describing the model of Hsc70 (HSPA8) promote anti-tumor immunity by targeting PD-L1 degradation through eMI: i) Hsc70 mediates the internalization of membrane-associated PD-L1; ii) Hsc70 facilitates the transport of cytosolic PD-L1 to late endosomes where it undergoes degradation by lysosomes; iii) Hsc70 competes with CMTM6 for binding sites on PD-L1, thus diminishing PD-L1 to recycle back to the cell membrane; iv) Augmentation in Hsc70 levels induced by AUY-922 can enhance PD-L1 internalization and degradation; v) Diminished recycling of PD-L1 reduces the interaction between PD-L1 on tumor cells and PD-1 on T cells, ultimately reactivating T cell-mediated anti-tumor immunity.

The graph abstract are shown as follows:

Figure 8 Schematic model for Hsc70 promoted anti-tumor immunity by targeting PD-L1 degradation through eMI

Minor comments

1. In the introduction, providing information about the functions and differences between CMA and eMI would help readers understand better.

Response: Thank you for the comments.

We have added the information about the functions and differences between CMA and eMI in the introduction which shown in red colors.

2. In Supplementary Fig. 1A, a masking effect could occur if the treated antibody and the antibody used in the FACS analysis recognize the same region of PD-L1. Information on whether the two antibodies recognize different epitopes of PD-L1 should be provided. Alternatively, performing live imaging using PD-L1 antibodies labeled with different fluorescent dyes in combination with a lysotracker may provide an answer.

Response: We appreciate the reviewer's valuable comments.

In our experiment, the blocking antibody was obtained from Bioxcell with clone number 10F.9G2™, while the detection antibody was sourced from Biolegend with clone number MIH6. We have revised this information in Methods of revised manuscript.

To further investigate whether these two clones recognize the same epitope, we incubated

IFN- γ -treated 4T1 cells with the anti-PD-L1 Ab (10F.9G2™) for varying durations, and the surface fluorescence intensity of PD-L1 was analyzed by flow cytometry using antibodies of the same clone (10F.9G2) or the MIH6 clone. As shown in Response Letter Fig 6, when using the 10F.9G2 clone for detection, PD-L1 intensity drops to the same level as IgG control within 1min. In contrast, when the MIH6 clone was used for detection, the fluorescence intensity of PD-L1 gradually decreased, with detectable value after 1-min of pre-incubation and baseline level after 5 min. The dynamics of PD-L1 intensity detected using the MIH6 clone is consistent with the rapid internalization of PD-L1 after antibody pre-incubation which showed that PD-L1 internalization occurred within 400 s (Li, Lim et al. 2018), suggesting that the binding of MIH6 clones was unlikely to be interfered by the 10F.9G2 clone.

The corresponding results are shown as follows:

Response Letter Fig 6 the antibody 10F.9G2 used for blocking does not compete for the binding of the detection antibody MIH6

4T1 cells were pretreated with 100 ng/ml IFN- γ for 48 h, then replaced with DMEM containing Anti-PD-L1 antibody (12.5 μ g/ml) for 0, 1, 5, 15 and 30 min, collect cells after cleaning with PBS, Flow cytometry staining using two PD-L1 monoclonal antibodies conjugated with PE.

3. Please pay attention to the labels so that one can intuitively understand the conditions and what is being measured by looking at the picture. ex) F1-f: Lamp2a => Lamp2a-frag, F1-j, SF3-D_Y-axis: MFI => PD-L1 MFI

Response: Thank you for the comment.

We have made corresponding modifications in the Figures and Supplementary Figures.

4. In Fig. 2E and 3A, the band signal looks saturated, so the change in PD-L1 is not clear.

It is needed to replace them with the same picture at a lower exposure.

Response:

Thank you for the comments.

We replaced a new set of data with low PD-L1 exposure (Figure 2E; Figure 3A) in the manuscript.

The results are as follows:

Figure 2E U18666A treatment prevented the degradation of PD-L1 induced by Hsc70

MCF-7 cells were transfected with Flag or Hsc70-Flag for 12 h, treated with or without U18666A (5 µg/mL) for another 12 h. PD-L1 levels were detected by western blotting with indicated antibodies.

Figure 3A Knocking down Hsc70 prevented PD-L1 degradation induced by knocking down CMTM6

MCF-7 cells were transfected with siRNA of Hsc70 and CMTM6 for 60 h, PD-L1 levels were detected by western blotting with indicated antibodies.

Response:

Mullock, B. M., N. A. Bright, C. W. Fearon, S. R. Gray and J. P. Luzio (1998). "Fusion of lysosomes with late endosomes produces a hybrid organelle of intermediate density and is NSF

dependent." J Cell Biol **140**(3): 591-601.

Reviewer #3 (Remarks to the Author): with expertise in heat shock proteins

This manuscript describes a very interesting study on the fate of PD-L1 that is targeted as a strategy to ameliorate cancer that has demonstrated great success. The therapeutic intervention is aimed at down-regulating the presence of PD-L1 from the surface of cancer cells. However, this therapeutic approach has some challenges due to the development of resistance due, in part, to the recycling of PD-L1 to the cell surface after internalization, avowing the degradation of the protein. Therefore, investigations directed at enhancing PD-L1 degradation are of great importance. Indeed, this study was aimed to understand and develop approaches to degrade PD-L1. Consequently, the study is of high relevance.

The authors found that Hsc70 (HSPA8), a constitutive member of the heat shock family of proteins, is associated with PD-L1 and could direct the receptor to the lysosome for degradation. The study is well performed using a variety of controls and techniques to validate their initial hypothesis. Thus, the investigation is solid and mechanistic. The main point is that HSPA8 competes with CMTM6, a natural partner of PD-L1, for binding to PD-L1. Due to the capacity of HSPA8 to recognize phosphatidylserine phospholipids (PS) present on the cytosolic side of cellular membranes, it may help in directing PD-L1 to the lysosome for breaking down.

Although the study is well performed, there are several aspects that should be addressed. If the authors decide to revise the manuscript, it is very likely that its impact will be enhanced.

Response:

We thank the reviewer for the appreciation of our study and the constructive comments. We have addressed the comments in detail below, we have also performed new experiments, added 46 new experimental panels to the manuscript and additional 18 new experiment panels in Response Letter. The text, newly added to the manuscript, is indicated in red. We hope that the reviewer finds our response satisfactory. Please note that the figure citations in our responses below refer to the new (post-revision) figures.

Specific comments:

1. Among all proteins identified by the mass spectrographic analysis (Fig. 1A), several other chaperones were present, including HSPA1 and HSP90. HSPA1 is abundant in cancer cells and has an overlapping function with HSPA8, particularly in its ability to bind to PS. The authors should explain the rationale for selecting HSPA8 over HSPA1.

Response:

We express our gratitude to the reviewer for their valuable feedback. We chose Hsc70

mainly based on the following considerations:

Firstly, we agree with the reviewer that some housekeeping genes like α -tubulin, β -tubulin or actin may show non-specific binding to PD-L1 due to their abundant expression in cell. Therefore, we excluded these proteins from the list.

Secondly, we used Kaplan-Meier Plotter to analyze several top-ranked proteins in PD-L1 mass spectrometry data, including HSPA1, EEF1A1, FUS, HSPA90AA1, HSPA90AB1 and HSPA8. As shown in **Supplementary Fig 1B**, among these proteins, only the high expression of HSPA8 is positively correlated with the overall survival rate of tumor patients.

Taking into account all the aforementioned considerations, we have chosen HSPA8 (also known as Hsc70) as the focal point for our research endeavors. We have now revised the description of how this gene was selected in the results and **Supplementary Fig 1B** of revised manuscript.

The corresponding results are shown as follows:

2. HSPA8 is one of the most abundant proteins in the cytosol, with a concentration of 107 molecules per cell. Could the authors explain why it is necessary to overexpress the protein for the effect on PD-L1? Are the constitutive levels not enough for targeting PD-L1?

Response:

Thank you for the comments.

HSPA8/Hsc70 is primarily involved in two types of autophagy: chaperone-mediated autophagy and endosomal microautophagy. The basal level of HSPA8 maintains the fundamental levels of both types of autophagy within the organism. Our research indicates that PD-L1 can be degraded through endosomal microautophagy, and HSPA8 sustains the membrane recycling of PD-L1. By overexpressing Hsc70 (HSPA8) in MCF-7 cells and extracting cytosolic and membrane proteins, our results show that overexpressing Hsc70 can further reduce the levels of PD-L1 in both the cytosol and on the membrane (Supplementary Fig 9A), suggesting that overexpressing Hsc70 disrupts the membrane recycling of PD-L1, making it more prone to degradation through the endosomal-lysosomal pathway. We have now included the information in Discussion.

The corresponding results are shown as follows:

Supplementary Fig 9A Hsc70 exists on the cell membrane, and overexpression of Hsc70 decrease the levels of PD-L1 in both the cytosol and on the membrane

MCF-7 cells were pretreated with IFN- γ for 48 h, then transfected with Flag, Hsc70-WT-Flag or Hsc70-3KA-Flag (CMV promoter) for 24 h, collect cells to extract cytoplasmic and membrane proteins, detected the protein levels of Tubulin, PD-L1, Flag by western blotting.

3. How much do the HSPA8 levels increase overexpression compared to basal levels? A simple Western blot may do the job.

Response:

We thank the reviewer for the constructive comments. To further investigate the effect of HSPA8 overexpression, we transfected HEK293T cells (low PD-L1 levels) with HA or Hsc70-HA for 24h. Western blotting was then performed using both HA and Hsc70 antibodies to detect the levels of exogenous Hsc70 (anti-HA) and total Hsc70 protein, respectively. As shown in Response Letter Fig 10, the results demonstrated a significant increase in Hsc70 (HSPA8) protein levels upon overexpression, compared to the baseline levels.

The corresponding results are shown as follows:

Response Letter Fig 10 overexpression of Hsc70 increased the levels of Hsc70

HEK293T cells were transfected with HA or Hsc70-HA (CMV promoter) for 24 h, cell lysates were immunoblotted with indicated antibodies.

4. The authors used a commercial polyclonal antibody to HSPA8. Are the authors sure that there is no cross-reaction with HSPA1? Maybe they should demonstrate the lack of cross-reaction using commercially available recombinant proteins for HSPA8 and HSPA1.

Response:

We are grateful for the reviewer's valuable feedback. To ensure the specificity of our antibody against HSPA8 (Hsc70) and rule out any potential cross-reaction with HSPA1, we conducted a series of experiments in MCF-7 cells.

We overexpressed HSPA1-Flag using the CMV promoter and observed a significant upregulation of both Flag and HSPA1 levels. Importantly, this did not affect the levels of Hsc70 (HSPA8) (Response Letter Fig 11). Additionally, we knocked down HSPA1 and observed a decrease in HSPA1 levels, without any change in Hsc70 (HSPA8) levels (Response Letter Fig 12). Results from these two assays together indicated no cross-reaction of anti-HSPA8 antibody against HSPA1.

The corresponding results are shown as follows:

Response Letter Fig 11 overexpression of HSPA1 increased the levels of HSPA1 but not influence the levels of Hsc70

MCF-7 cells were transfected with Flag of HSPA1-Flag (CMV promoter) for 24 h, cell lysates were immunoblotted by indicated antibodies.

Response Letter Fig 12 knockdown of HSPA1 decreased the levels of HSPA1 but not influence the levels of Hsc70 (HSPA8)

MCF-7 cells were transfected with siRNA of HSPA1 for 60 h, cell lysates were immunoblotted by indicated antibodies.

5. The reviewer is concerned that overexpression of HSPA8 may be toxic for the cell, as it has been reported in the past (see Feder et al. 1992, Genes Dev 6:1402–1413; Arispe et al. 2004, FASEB J 18:1636–1645). Did the authors check cellular viability after overexpression of HSPA8?

Response:

We appreciate the reviewer's comments, which have helped us further clarify our findings. In this study, we overexpressed Hsc70 (HSPA8) in both HEK293T (human embryonic kidney) and MCF-7 (human breast cancer) cells to investigate its impact on cell viability. Our results showed that while overexpression of Hsc70 had no significant effect on cell viability in both HEK293T cells and MCF-7 cells (Response Letter Fig 13).

The corresponding results are shown as follows:

Response Letter Fig 13 Overexpression of Hsc70 have no effect on the cell viability of HEK293T cells and MCF-7 cells

HEK293T cells and MCF-7 cells were transfected with Flag or Hsc70-Flag (CMV promoter) for 24 h, collect cells and inoculate into 96 wells with 1×10^4 /well for 24 h, cell viability was determined using CellTiterGlo® assay (data represents Mean \pm SD; **** $p < 0.0001$, t-test).

6. The authors concluded that HSPA8 and CMTM6 bind to the same side on PD-L1, within aa 161-241. Would the authors indicate that the region is in the PD-L1, cytosolic, or membrane domain?

Response:

Thank you for your comments. The aa 161-241 segment corresponds to the extracellular portion of the cell membrane PD-L1 protein.

7. Mechanistically, the authors assume that HSPA8 is binding to PD-L1, perhaps at the cytosolic side (not mentioned in the manuscript), and target the endocytic vesicle to the lysosome, avoiding receptors recycling. If the reviewer got this concept right, could the authors explain or speculate the mechanism involved? Is HSPA8 interacting with a specific component of lysosomes? The authors may like to check the manuscript by Kirkegaard et al. 2010, Nature 463:549–553, and Mahalka et al. 2014, Biochim Biophys Acta 1838:1344–1361, which showed the interaction of HSPA1 with the negatively charged phospholipid BMP, which is a major component of the lysosome membrane. The authors may also like to check an excellent review of the interaction of HSP with membranes (De Maio and Hightower 2021, Cell Stress Chaperones 26:769-783).

Response:

Thank you for your insightful comments. HSPA8 (Hsc70) plays distinct roles in two types of selective autophagy: chaperone-mediated autophagy (CMA) and endosomal microautophagy (eMI). In CMA, HSPA8 interacts with Lamp2a, which is localized in lysosomes and facilitates the unfolding of substrate proteins and their subsequent delivery to lysosomes for degradation.

In our study, the mechanism of HSPA8 regulating PD-L1 degradation was elucidated mainly through two sets of assays. First, as shown in Figure 1E-J; Supplementary Fig 1H-M, PD-L1 degradation induced by Hsc70 is dependent on lysosome but not on CMA. Second, we used Hsc70-3KA, an Hsc70 mutant form on its C-terminal lysine residues which disrupts the binding to phosphatidylserine (Sahu, Kaushik et al. 2011), showing in Supplementary Fig 9A that overexpression of Hsc70-3KA failed to decrease the protein levels of PD-L1 on the membrane and in cytoplasmic. These results were also consistent the previous reports that the binding of HSPA8 carboxy-terminal lid domain with phosphatidylserine on the late endosome membrane is essential for eMI (Yeung, Gilbert et al. 2008, Morozova, Clement et al. 2016).

We do not rule out that HSPA1 exhibits high selectivity towards negatively charged phospholipids. However, Hsc70/HSPA8 also exhibit high selectivity towards negatively charged phospholipids like phosphatidylserine and HSPA8 is capable of inserting into the lipid bilayer (Arispe and De Maio 2000, Dores-Silva, Cauvi et al. 2021). HSPA8 also has been reported to be expressed on the surface of certain tumor cells (Mills, Haskell et al. 2010) and has been detected on endosome membranes participating in the microautophagy process (Sahu, Kaushik et al. 2011, Morozova, Clement et al. 2016, Dores-Silva, Cauvi et al. 2021). Our findings align with these observations. As shown in Supplementary Fig 9A, B, Hsc70 exists not only in the cytoplasmic but also on membrane.

Therefore, we infer that endogenous-level Hsc70 plays a critical role in mediating basal levels of PD-L1 recycling and degradation. In contrast, an elevated expression of Hsc70 or an augmentation in Hsc70 levels induced by AUY-922 disrupts the fundamental PD-L1 recycling mechanism. On one hand, this results in enhanced internalization of cytoplasmic PD-L1 by Hsc70, leading to its transport to late endosomes where it undergoes degradation by lysosomes. On the other hand, Hsc70 competes with CMTM6 for binding sites on PD-L1, thus diminishing PD-L1 to recycle back to the cell membrane via CMTM6. Together, this diminished recycling reduces the interaction between PD-L1 on tumor cells and PD-1 on T cells, ultimately reactivating T cell-mediated anti-tumor immunity.

The corresponding results are shown as follows:

Supplementary Fig 9A Hsc70 and PD-L1 exists not only on the membrane but also in the cytoplasmic, and overexpression of Hsc70 promoted the degradation of PD-L1 on the membrane and in the cytoplasmic

MCF-7 cells were pretreated with IFN- γ for 48 h, then transfected with Flag, Hsc70-WT-Flag or Hsc70-3KA-Flag (CMV promoter) for 24 h, collect cells to extract cytoplasmic and membrane proteins, detected the protein levels of Tubulin, PD-L1, Hsc70, Flag by western blotting.

Supplementary Fig 9B PD-L1 exists on the cell membrane and in the cytoplasmic, AUY-922 treatment promoted the degradation of PD-L1 on the membrane and in the cytoplasmic

Supplementary Fig 9B MCF-7 cells were pretreated with IFN- γ for 48 h, then treated with or without AUY-922 for 12 h, collect cells to extract cytoplasmic and membrane proteins, detected the protein levels of Tubulin, PD-L1, Hsc70 by western blotting.

8. The reviewer is concerned that the observations from this study on the involvement of HSPA8 in vesicular traffic are solely due to this protein, or perhaps HSPA1 is also playing a role. HSPA1 is very abundant in cancer cells but not in normal tissues. Therefore, it may be a more appropriate target.

Response:

Thank you for your comments. While we acknowledge that HSPA1 may be a potential target, we also realize that HSPA1 and HSPA8 are distinct targets with distinct functions. Previous studies have shown that HSPA1 is essential for tumor growth and survival (Nylandsted, Brand et al. 2000). Our results also showed that overexpression of HSPA1 can increase the cell viability of HEK293T and MCF-7 cells (Response Letter Fig 14). This excessive proliferation of normal cells highlights the potential oncogenic risk associated with HSPA1. Furthermore, Kaplan-Meier Plotter analysis reveals that tumor patients expressing high levels of HSPA1 exhibit shorter overall survival (Response Letter Fig 15).

These results indicate a positive correlation between HSPA1 and tumor growth.

In contrast, our results show that overexpression of HSPA8 does not impact the cell viability of HEK293T cells and MCF-7 cells (Response Letter Fig 13). Kaplan-Meier Plotter analysis revealing that tumor patients expressing high levels of HSPA8 have a longer overall survival (Response Letter Fig 16). These findings indicate a negative correlation between HSPA8 and tumor growth.

Based on the above considerations, we have chosen HSPA8 as the research object.

The corresponding results are as follows:

Response Letter Fig 14 overexpression of HSPA1 increased the cell viability of HEK293T and MCF-7

HEK293T cells and MCF-7 cells were transfected with Flag or HSPA1A-Flag (CMV promoter) for 24 h, collect cells and inoculate into 96 wells with 1×10^4 /well for 24 h, cell viability was determined using CellTiterGlo® assay (data represents Mean \pm SD; ****p<0.0001, t-test).

Response Letter Fig 15 Tumor patients with high expression of HSPA1 had a shorter survival period

Correlation analysis between HSPA1A gene levels with overall survival (OS) in all tumor types of patients who have not received any immune checkpoint inhibitors (ICIs) treatment from the Kaplan-Meier Plotter platform ([Kaplan-Meier plotter \[Immunotherapy\] \(kmplot.com\)](https://www.kmplot.com)).

Response Letter Fig 13 Overexpression of Hsc70 does not affect the cell viability of HEK293T cells and MCF-7 cells

HEK293T cells and MCF-7 cells were transfected with Flag or Hsc70-Flag (CMV promoter) for 24 h, collect cells and inoculate into 96 wells with 1×10^4 /well for 24 h, cell viability was determined using CellTiterGlo® assay (data represents Mean \pm SD; **** $p < 0.0001$, t-test).

Response Letter Fig 16 Tumor patients with high expression of HSPA8 had a longer survival period

Correlation analysis between HSPA8 gene levels with overall survival (OS) in all tumor types of patients who have not received any immune checkpoint inhibitors (ICIs) treatment from the Kaplan-Meier Plotter platform ([Kaplan-Meier plotter \[Immunotherapy\] \(kmplot.com\)](https://www.kmplot.com)).

9. The authors have shown that the HSP90 inhibitor AUY-922 increases the expression of HSPA8. However, it is well established that these inhibitors elevate the expression of HSPA1 by increasing the transcriptional activity of HSF1. Moreover, increasing the expression of HSPA1 by stress or by HSP90 inhibitors resulted in an increase in the endocytic/vesicular process (see Vega et al. 2010 Cell Stress Chaperones 15:517–527). There are not prior reports, to the best of the reviewer, that HSPA8 expression increases upon inhibition of HSP90. These observations may indeed point to the direction of a possible role of HSPA1.

Response:

Thank you for the comments.

We have now evaluated HSPA1 and HSPA8 expression in response to two HSP90 inhibitors (AUY-922 and 17-AAG), confirming that HSPA1 and HSPA8 all are upregulated after HSP90 inhibitor treatment at the transcriptional level (Response Letter Fig 17) and protein levels (Figure 6B, Supplementary Fig 6A-C, Response Letter Fig 18). However, overexpression or knockdown of HSPA1 did not influence the protein levels of HSPA8 and PD-L1 (Response Letter Fig 11-12). Due to their shared membership in the HSP70 family and similarities in function and structure, endogenously expressed HSPA8 and stress-induced HSPA1 may exhibit some degree of functional redundancy.

Given the constructive expression of HSPA8 and its association with longer median survival in patients, we have chosen it as the focus of our research.

The corresponding results are as follows:

Response Letter Fig 17 HSP90 inhibitors treatment increased the mRNA levels of HSPA1 and HSPA8 (Hsc70)

MCF-7 cells were treated with 1 μ M AUY-922 or 17-AAG for 12 h, the mRNA levels of HSPA1 and Hsc70 (HSPA8) were analyzed by Real-time PCR.

Response Letter Fig 18 AUY-922 treatment increased the protein level of HSPA1

MCF-7 cells were treated with AUY-922 (1 μ M) for 12 h, cell lysates were immunoblotted by indicated antibodies.

Response Letter Fig 11 Overexpression of HSPA1 did not influence the levels of Hsc70 and PD-L1

MCF-7 cells were transfected with Flag or HSPA1-Flag (CMV promoter) for 24 h, cell lysates were immunoblotted by indicated antibodies.

Response Letter Fig 12 Knockdown of HSPA1 did not influence the levels of Hsc70 (HSPA8) and PD-L1

MCF-7 cells were transfected with siRNA of HSPA1 for 60 h, cell lysates were immunoblotted by indicated antibodies.

10. The manuscript will be greatly enhanced if a model (even speculative) of the process of HSPA8 interaction with PD-L1 and targeting the lysosome is presented. It may be a great resource for experts and non-experts on the subject.

Response:

Thank you for the comments. We have now added **Figure 8** describing the model of how Hsc70 (HSPA8) regulates PD-L1 expression and promoted anti-tumor immunity: i) HSPA8 mediates the internalization of membrane-associated PD-L1; ii) HSPA8 facilitates the transport of cytosolic PD-L1 to late endosomes where it undergoes degradation by lysosomes; iii) HSPA8 competes with CMTM6 for binding sites on PD-L1, thus diminishing PD-L1 to recycle back to the cell membrane; iv) Augmentation in HSPA8 levels induced by AUY-922 can enhance PD-L1 internalization and degradation; v) Diminished recycling of PD-L1 reduces the interaction between PD-L1 on tumor cells and PD-1 on T cells, ultimately reactivating T cell-mediated anti-tumor immunity.

The graph abstract are shown as follows:

Reference:

Arispe, N. and A. De Maio (2000). "ATP and ADP modulate a cation channel formed by Hsc70 in acidic phospholipid membranes." *J Biol Chem* 275(40): 30839-30843.

Dores-Silva, P. R., D. M. Cauvi, A. L. S. Coto, N. S. M. Silva, J. C. Borges and A. De Maio (2021). "Human heat shock cognate protein (HSC70/HSPA8) interacts with negatively charged phospholipids by a different mechanism than other HSP70s and brings HSP90 into

membranes." Cell Stress Chaperones **26**(4): 671-684.

Mills, D. R., M. D. Haskell, H. M. Callanan, D. L. Flanagan, K. E. Brilliant, D. Yang and D. C. Hixson (2010). "Monoclonal antibody to novel cell surface epitope on Hsc70 promotes morphogenesis of bile ducts in newborn rat liver." Cell Stress Chaperones **15**(1): 39-53.

Morozova, K., C. C. Clement, S. Kaushik, B. Stiller, E. Arias, A. Ahmad, J. N. Rauch, V. Chatterjee, C. Melis, B. Scharf, J. E. Gestwicki, A. M. Cuervo, E. R. Zuiderweg and L. Santambrogio (2016). "Structural and Biological Interaction of hsc-70 Protein with Phosphatidylserine in Endosomal Microautophagy." J Biol Chem **291**(35): 18096-18106.

Nylandsted, J., K. Brand and M. Jaattela (2000). "Heat shock protein 70 is required for the survival of cancer cells." Ann N Y Acad Sci **926**: 122-125.

Sahu, R., S. Kaushik, C. C. Clement, E. S. Cannizzo, B. Scharf, A. Follenzi, I. Potolicchio, E. Nieves, A. M. Cuervo and L. Santambrogio (2011). "Microautophagy of cytosolic proteins by late endosomes." Dev Cell **20**(1): 131-139.

Yeung, T., G. E. Gilbert, J. Shi, J. Silvius, A. Kapus and S. Grinstein (2008). "Membrane phosphatidylserine regulates surface charge and protein localization." Science **319**(5860): 210-213.

Reviewer #4 (Remarks to the Author): with expertise in heat shock proteins, immunology

Paper titled "Hsc70 promotes anti-tumor immunity by targeting PD-L1 for lysosomal degradation" identifies new molecular mechanism of Hsc70-mediated PDL-1 lysosomal degradation and provides new approach to improve efficacy of checkpoint inhibition therapy (PD-1/PD-L1). As pointed by the authors, just very recently was uncovered that after aPD-L1 treatment, PD-L1 is re-expressed on the surface of the cell and this mechanism confers resistance to ICI therapy. Hence, controlling the re-expression of PD-L1 is very important and this paper provides new avenues for reducing the amount of PD-L1 recirculating back to the cell membrane. This work presents very original findings that Hsp90 inhibitor, AUY922 facilitates PD-L1 degradation via lysosomes and boost anti-tumor immune responses by augmenting Hsc70- expression.

Methodology used is appropriate and provides enough details for this work to be reproduced.

Response:

We thank the reviewer for the appreciation of our study and the constructive comments. We have addressed the comments in detail below, we have also performed new experiments, added 46 new experimental panels to the manuscript and additional 18 new experiment panels in Response Letter. The text, newly added to the manuscript, is indicated in red. We hope that the reviewer finds our response satisfactory. Please note that the figure citations in our responses below refer to the new (post-revision) figures.

I do not have any major, just a few minor concerns.

1. Authors should include summary graph for all data showing representative surface expression of PD-L1 by flow cytometry (Fig 1C and H; Fig 2B and D, F and H; Fig 3B and Fig 7A)

Response:

Thank you for the constructive comments, As shown in **Supplementary Fig1D, 1J, 2C, 2D, 2F, 2H, 3E, 6R**, we have added the corresponding statistical data for flow analysis in the manuscript.

The results are as follows:

Supplementary Fig 1D

MCF-7 cells were transfected with 1 µg and 2 µg Hsc70-Flag for 24 h, fluorescence of PD-L1 on the surface of cell membrane was analyzed by flow cytometry.

Supplementary Fig 1J

MCF-7 cells were transfected with siRNAs of Lamp2a for 36 h, transfected with or without Hsc70-Flag for another 24 h, fluorescence of PD-L1 on the surface of cell membrane was analyzed by flow cytometry.

Supplementary Fig 2C

MCF-7 cells were transfected with siRNAs of TSG101 for 36 h, transfected with or without Hsc70-Flag for another 24 h. Fluorescence of PD-L1 on the surface of cell membrane was analyzed by flow cytometry.

Supplementary Fig 2D

MCF-7 cells were transfected with siRNAs of VPS4 for 36 h, transfected with or without Hsc70-Flag for another 24 h. Fluorescence of PD-L1 on the surface of cell membrane was analyzed by flow cytometry.

Supplementary Fig 2F

MCF-7 cells were transfected with Flag or Hsc70-Flag for 12 h, treated with or without U18666A (5 μ g/mL) for another 12 h. Fluorescence of PD-L1 on the surface of cell membrane was analyzed by flow cytometry.

Supplementary Fig 2H

MCF-7 cells were transfected with Hsc70-WT-Flag or Hsc70-3KA-Flag plasmids for 24 h. Fluorescence of PD-L1 on the surface of cell membrane was analyzed by flow cytometry.

Supplementary Fig 3E

MCF7 cells were transfected with siRNA of Hsc70 and CMTM6 for 60 h, fluorescence of PD-L1 on the surface of cell membrane was analyzed by flow cytometry.

Supplementary Fig 6R

4T1 cells were pretreated with IFN- γ (100 ng/mL) and treated with Anti-PD-L1 Ab (100 nM) for 24 h or 96 h, then treated with or without 1 μ M AUY-922 for another 24 h, fluorescence of PD-L1 on the surface of cell membrane was analyzed by flow cytometry.

2. Authors should state that reported MFI values on y-axes' are PD-L1 MFI throughout the

manuscript (Fig 1F and J; Fig 3 E, G and I; Fig 4 H; Fig 6 E, G, H and I; Fig 7 H).

Response:

Thank you for the comments. We have changed the MFI of the Y-axis to PD-L1 MFI in the Figures of revised manuscript.

3. In Fig 7 F and H there is no additive effect on the frequency of CD8+ T cells or Gzmb+ CD8+ T cells after combination of AUY-922 + anti-PD-L1 (there is no statistical significance between AUY-922 and AUY-922+anti-PD-L1). How do you explain this (since there is statistically significant effect on the tumor volume and weight). Also, in H, there is no significant difference on the surface expression levels of Pd-L1 between AUY and AUY+anti-PD-L1.

Response:

Thank you for your comments. In this experiment, anti-PD-L1 was administered once weekly. Previous research and our findings in 4T1 cells indicate that anti-PD-L1 treatment rapidly reduces PD-L1 levels, but over extended periods, internalized PD-L1 is recycled back to the cell membrane via recycling endosomes. Our final assessment occurred 7 days post-anti-PD-L1 administration, a time point at which PD-L1 levels had already recuperated. Consequently, there was no notable difference in PD-L1 expression between treatments with AUY-922 alone and the combination of AUY-922 and anti-PD-L1. Additionally, we conducted western blotting to measure PD-L1 levels in harvested tumor cells, confirming that both AUY-922 and the combined therapy reduced PD-L1 levels.

Regarding CD8 and GzmB, we also employed immunohistochemistry for detection. The results revealed that both AUY-922 and anti-PD-L1 monotherapy significantly increased the levels of CD8 and GzmB, while the combination of AUY-922 and anti-PD-L1 further elevated the levels of CD8 and GZMB. The immunohistochemical findings serve as a complement to the flow cytometry results.

The corresponding results are shown as follows:

Figure 7H AUY-922 or the combined of AUY-922 and anti-PD-L1 decreased the levels of PD-L1 and increased the levels of Hsc70

Indicated protein levels were detected by Immunoblotting with the indicated harvested tumor cells.

Figure 7J Immunohistochemistry analysis showed a significant elevation of CD8 and GzmB in the combined of AUY-922 and anti-PD-L1

Immunohistochemistry analysis of CD8 and GzmB in the 4T1 tumors. Scale bar, 50 μ m. n=15 (each group has 5 tumor tissues, and each tissue contain 3 random fields).

4. Legend of Fig 5: CD8 is not a CTL marker, should be replaced with surface marker

Response:

Thank you for pointing this detail out. We have replaced CD8 a CTL marker as the surface marker.

Overall, I recommend this paper for publication.

REVIEWER COMMENTS

Reviewer #1 (Remarks to the Author):

1. Supplementary Fig 5. A detailed method for preparing the tumor cell lysate should be provided to clarify the procedure. It is essential to specify whether the lysate originated from pure tumor cells, as indicated in the legend, or if it was a mixture of tumor cells and other cells within the tumor block. Additionally, the authors should elaborate on how sample selection was conducted for inclusion in the Western blot analysis.

2. Previous comment “3. There is no evidence of a reduction in PD-L1 in Hsc70 WT overexpression cells compared to Flag control cells in (original version) Figure 2I, 2J, 2K, 6I, 6J, nor any reduction of PD-L1 in CMTM6 knockdown cells in (original version) Figure 3J, 3K, and 3L. However, Western blot data show a clear reduction in both cases. Could the authors explain this discrepancy?”

This comment has not been convincingly addressed. The authors aim to demonstrate the effects of Hsc70 and CMTM6 on the subcellular localization of PD-L1. It is unclear why the authors chose to include early and various time points, where the ectopically introduced proteins may not have commenced their functional activity, for the immunofluorescent experiment. Conducting the IF experiment with cells treated under the same conditions as those for the western blot would provide a more relevant comparison.

3. Previous comment “7. Supplementary Figure 1A and Figure 7A: Is there any evidence that the antibody used for blocking does not compete for the binding of the detection antibody? The methods used for these experiments should be described in more detail.”

This comment remains inadequately addressed. The results presented in Response Letter Fig 6 indicate significant influence on both MIH6 and 10F.9G2 antibody clones by the blocking antibody. Some PD-L1 proteins may get internalized, but PD-L1 is also produced or recycled in the same time. The detection of the PD-L1 at the end point was not properly done.

To quantify PD-L1 in Figure 7A, I recommend that the authors use the same blocking antibody at the same concentration as used for blocking to stain the cells. Subsequently, they can employ a fluorophore-conjugated secondary antibody for quantification via flow cytometry.

Reviewer #2 (Remarks to the Author):

The authors have satisfactorily addressed most of the reviewer's comments through additional explanations and experiments, and the quality of the manuscript has significantly improved after revisions. There is no dispute that HSC70 (HSPA8) is crucial for the regulation of PD-L1 expression and that targeting HSC70 can be potentially combined with anticancer immunotherapy.

However, the response to Major Comment 1 is still confusing and requires clarification. A clear rationale for choosing HSC70 (HSPA8) as the target of study is needed to strengthen the validity of the study, as several chaperone proteins with non-specific binding potential were detected due to the high intracellular expression levels identified in the mass ID analysis. Reviewer #3 raised similar concerns.

The main response to major comment 1 was that among the candidate proteins, only HSPA8 showed a positive association with overall survival (OS). Despite the availability of pan-cancer RNA-seq database (n=7462), the authors chose to base their analysis on patients who received immunotherapy (n=972). Is there a specific reason for this? Additionally, using an immunotherapy-based analysis while examining patients who have never received immune checkpoint inhibitors (ICIs) for OS analysis is puzzling. Further explanation is needed to fully address major comment 1.

Reviewer #3 (Remarks to the Author):

The authors have answer my questions and concerns

Reviewer #4 (Remarks to the Author):

All concerns have been addresses. I do not have any additional questions. Thank you

REVIEWER COMMENTS

Reviewer #1 (Remarks to the Author):

1. Supplementary Fig 5. A detailed method for preparing the tumor cell lysate should be provided to clarify the procedure. It is essential to specify whether the lysate originated from pure tumor cells, as indicated in the legend, or if it was a mixture of tumor cells and other cells within the tumor block. Additionally, the authors should elaborate on how sample selection was conducted for inclusion in the Western blot analysis.

Response:

Thank you for the comments.

For immunoblotting, after digesting tumor tissues with collagenase type I (#2350118, Gibco) and DNase I (#143582, Roche), and filtering the resulting cells through 45 µm filter, 100 µL of total cells suspension is extracted. An equal volume of 2×loading buffer is then added, and the mixture is boiled at 100°C for 10 min to prepare the total tumor lysates. The tumor tissue samples (n=3) included in the western blotting analysis were randomly selected. The detailed methods are shown in the **methods** of revised manuscript.

Given the fact that total tumor lysates were used, we are aware that the immune cells within the tumor may have an impact on evaluation of PD-L1 levels of tumor cells. Therefore, we carefully revisited all flow cytometry data. As shown in **Response Letter Figure 1**, there are no statistically significant differences in the proportion of CD45⁺ cells across all tumor tissue samples.

The results are shown in the follows:

Response Letter Figure 1 There is no significant difference in the proportion of CD45 ⁺ immune cells in the total cells
--

2. Previous comment “3. There is no evidence of a reduction in PD-L1 in Hsc70 WT overexpression cells compared to Flag control cells in (original version) Figure 2I, 2J, 2K, 6I, 6J, nor any reduction of PD-L1 in CMTM6 knockdown cells in (original version) Figure 3J, 3K, and 3L. However, Western blot data show a clear reduction in both cases. Could the authors explain this discrepancy?”

This comment has not been convincingly addressed. The authors aim to demonstrate the effects of Hsc70 and CMTM6 on the subcellular localization of PD-L1. It is unclear why the authors chose to include early and various time points, where the ectopically introduced proteins may not have commenced their functional activity, for the immunofluorescent experiment. Conducting the IF experiment with cells treated under the same conditions as those for the western blot would provide a more relevant comparison.

Response:

Thank you for the comments.

For immunofluorescent assays, we specifically selected these time points when PD-L1 had not undergone significant degradation. These time points enabled us to observe the co-localization of PD-L1 with specific organelles during its degradation process. Conversely, selecting time points when PD-L1 degradation was prominent would have hindered our ability to accurately observe its interactions with cellular organelles, given the marked decrease in PD-L1 levels. This approach ensures that any observed interactions are not confounded by the degradation process, allowing us to accurately assess the association between two proteins. We have revised the description in Methods for better clarification.

3. Previous comment “7. Supplementary Figure 1A and Figure 7A: Is there any evidence that the antibody used for blocking does not compete for the binding of the detection antibody? The methods used for these experiments should be described in more detail.”

This comment remains inadequately addressed. The results presented in Response Letter Fig 6 indicate significant influence on both MIH6 and 10F.9G2 antibody clones by the blocking antibody. Some PD-L1 proteins may get internalized, but PD-L1 is also produced or recycled in the same time. The detection of the PD-L1 at the end point was not properly done.

To quantify PD-L1 in Figure 7A, I recommend that the authors use the same blocking antibody at the same concentration as used for blocking to stain the cells. Subsequently, they can employ a fluorophore-conjugated secondary antibody for quantification via flow cytometry.

Response:

Thank you for the comments.

Based on your suggestion, we used anti-PD-L1 Ab (#BE0101, Bioxcell) as the primary antibody for blocking, and then stained with fluorescent secondary antibody PE anti-rat IgG2b (#408214, Biolegend). Finally, the levels of PD-L1 on the cell membrane were detected by flow cytometry.

As shown in **Supplementary Figure 8**, after treatment with PD-L1 monoclonal antibody for 24h, the fluorescence of PD-L1 on the cell membrane decreased significantly. After 96h, the fluorescence of PD-L1 recovered. On this basis, another 24h of treatment with AUY-922 further down-regulate the fluorescence of PD-L1 on cell membrane.

We have also validated this experiment using Western blotting. As shown in **Response Letter Figure 2**, Western blotting analysis showed that treatment with anti-PD-L1 Ab for 24h, downregulated the level of PD-L1, but for 96h, the levels of PD-L1 rebounded. However, concurrent treatment with AUY-922 for 24h was able to re-downregulate the levels of PD-L1.

The results are shown in the follows, and we have replaced the flow cytometry data in the Supplementary Fig 8.

Supplementary Figure 8 after 24h of PD-L1 monoclonal antibody treatment, membrane PD-L1 levels significantly decreased, but recovered at 96h, however, subsequent AUY-922 treatment effectively downregulated PD-L1 levels

4T1 cells were pretreated with IFN- γ (100 ng/mL) and treated with Anti-PD-L1 Ab (100 nM) for 24 h or 96 h, then treated with or without 1 μ M AUY-922 for another 24 h. PD-L1 monoclonal antibody was used as the primary antibody for blocking, followed by fluorescent secondary antibody staining. PD-L1 levels were detected by flow cytometry.

Response Letter Figure 2 Western blotting results showed that prolonged treatment with anti-PD-L1 Ab restores the level of PD-L1, while treatment with AUY-922 reduced it

4T1 cells were pretreated with IFN- γ (100 ng/mL) and treated with Anti-PD-L1 Ab (100 nM) for 24 h or 96 h, then treated with or without 1 μ M AUY-922 for another 24 h, cell lysates were immunoblotted with indicated antibodies.

Reviewer #2 (Remarks to the Author):

The authors have satisfactorily addressed most of the reviewer's comments through additional explanations and experiments, and the quality of the manuscript has significantly improved after revisions. There is no dispute that HSC70 (HSPA8) is crucial for the regulation of PD-L1 expression and that targeting HSC70 can be potentially combined with anticancer immunotherapy.

However, the response to Major Comment 1 is still confusing and requires clarification. A clear rationale for choosing HSC70 (HSPA8) as the target of study is needed to strengthen the validity of the study, as several chaperone proteins with non-specific binding potential were detected due to the high intracellular expression levels identified in the mass ID analysis. Reviewer #3 raised similar concerns.

The main response to major comment 1 was that among the candidate proteins, only HSPA8 showed a positive association with overall survival (OS). Despite the availability of pan-cancer RNA-seq database (n=7462), the authors chose to base their analysis on patients who received immunotherapy (n=972). Is there a specific reason for this? Additionally, using an immunotherapy-based analysis while examining patients who have never received immune checkpoint inhibitors (ICIs) for OS analysis is puzzling. Further explanation is needed to fully address major comment 1.

Response:

Thank you for the comments, and we have revised these analyses to better support the rationale for selecting HSPA8 (Hsc70) in our study.

First, we revised the Kaplan-Meier Plotter analysis of the protein levels with overall survival (OS) in breast cancer patients (Kaplan-Meier plotter [proteomics] (kmplot.com)). In the high confidence interacting proteins, only HSPA8 and FUS were included in this database, so we sought to analyze these two proteins. As shown in **Supplementary Figure 1B**, these two protein levels are positively correlated with the overall survival rate of tumor patients. These results provide evidence that Hsc70 expression is correlated with the aggressiveness of breast cancer which was further elaborated functionally and mechanistically in our study on breast cancer cells and animal models.

Second, we revised the immunotherapy-related analyses to include patients receiving anti-PD-L1 immunotherapy agents Kaplan-Meier plotter [Immunotherapy] (kmplot.com). As shown in **Supplementary Figure 1C**, HSPA8 was positively correlated with OS in anti-PD-L1 immunotherapy, however, FUS was negatively correlated with OS in anti-PD-L1 immunotherapy.

Taking all the above considerations, we have chosen HSPA8 as the focal point for our research endeavors.

The results are shown as follows:

Supplementary Figure 1B Only HSPA8 and FUS protein levels are included in the database of breast protein

Correlation analysis between HSPA8 and FUS protein levels with overall survival (OS) in breast cancer patients from the Kaplan-Meier Plotter platform ([Kaplan-Meier plotter \[proteomics\] \(kmplot.com\)](https://www.kmplot.com))

Supplementary Figure 1C HSPA8 expression is positively correlated with OS in all tumor types of patients who have received anti-PD-L1 immunotherapy

Correlation analysis between HSPA8 and FUS protein levels with overall survival (OS) in all tumor types of patients who have received anti-PD-L1 immune checkpoint inhibitors treatment from the Kaplan-Meier Plotter platform ([Kaplan-Meier plotter \[proteomics\] \(kmplot.com\)](https://www.kmplot.com))

Reviewer #3 (Remarks to the Author):

The authors have answer my questions and concerns

Response:

Thank you for the comments.

Reviewer #4 (Remarks to the Author):

All concerns have been addresses. I do not have any additional questions. Thank you

Response:

Thank you for the comments.

REVIEWERS' COMMENTS

Reviewer #1 (Remarks to the Author):

I do not have further comments.

Reviewer #2 (Remarks to the Author):

The authors have answered my questions and concerns. I appreciate their thorough responses.

REVIEWERS' COMMENTS

Reviewer #1 (Remarks to the Author):

I do not have further comments.

Response:

We thank the reviewer for their appreciation of our study and the constructive comments.

Reviewer #2 (Remarks to the Author):

The authors have answered my questions and concerns. I appreciate their thorough responses.

Response:

We thank the reviewer for their appreciation of our study and the constructive comments.